# Self-Improving Embodied Foundation Models

Seyed **Kamyar** Seyed **Ghasemipour** *
Generalist AI
kamyar@generalistai.com

**Ayzaan Wahid & Jonathan Tompson & Pannag Sanketi[†] & Igor Mordatch[†]**
Google DeepMind
{ayzaan, tompson, psanketi, imordatch}@google.com

## Abstract

Foundation models trained on web-scale data have revolutionized robotics, but their application to low-level control remains largely limited to behavioral cloning. Drawing inspiration from the success of the reinforcement learning stage in fine-tuning large language models, we propose a two-stage post-training approach for robotics. The first stage, Supervised Fine-Tuning (SFT), fine-tunes pretrained foundation models using both: a) behavioral cloning, and b) steps-to-go prediction objectives. In the second stage, Self-Improvement, steps-to-go prediction enables the extraction of a well-shaped reward function and a robust success detector, enabling a fleet of robots to autonomously practice downstream tasks with minimal human supervision. Through extensive experiments on real-world and simulated robot embodiments, our novel post-training recipe unveils significant results on Embodied Foundation Models. First, we demonstrate that the combination of SFT and Self-Improvement is significantly more sample-efficient than scaling imitation data collection for supervised learning, and that it leads to policies with significantly higher success rates. Further ablations highlight that the combination of web-scale pretraining and Self-Improvement is the key to this sample-efficiency. Next, we demonstrate that our proposed combination uniquely unlocks a capability that current methods cannot achieve: autonomously practicing and acquiring novel skills that generalize far beyond the behaviors observed in the imitation learning datasets used during training. These findings highlight the transformative potential of combining pretrained foundation models with online Self-Improvement to enable autonomous skill acquisition in robotics.

## 1   Introduction

Recent works have demonstrated that foundation models can be effectively fine-tuned to directly act as low-level robot policies [9, 41, 45, 39, 31, 21, 7], and that they inherit significant generalization and robustness capabilities due to the web-scale pretraining of the foundation models from which they were derived. Thus far the training regime for Embodied Foundation Models (EFMs) has been limited to behavioral cloning (i.e. supervised learning) [9, 41, 45, 39, 31, 21, 7]. In contrast, from the literature on large language models (LLMs) we observe that after the initial pretraining, post-training for downstream tasks is typically divided into two stages: 1) Supervised Fine-Tuning (SFT), followed by 2) Reinforcement Learning (RL). RL-tuning of LLMs has been shown to markedly, and rapidly, improve downstream task performance beyond SFT [47, 40], and has become a critical stage in the training recipe of foundation models [25, 1, 50, 20].

---

*Founding Member of Technical Staff at Generalist AI. Project completed April 2024 at Google DeepMind.
[†]Equal supervision.

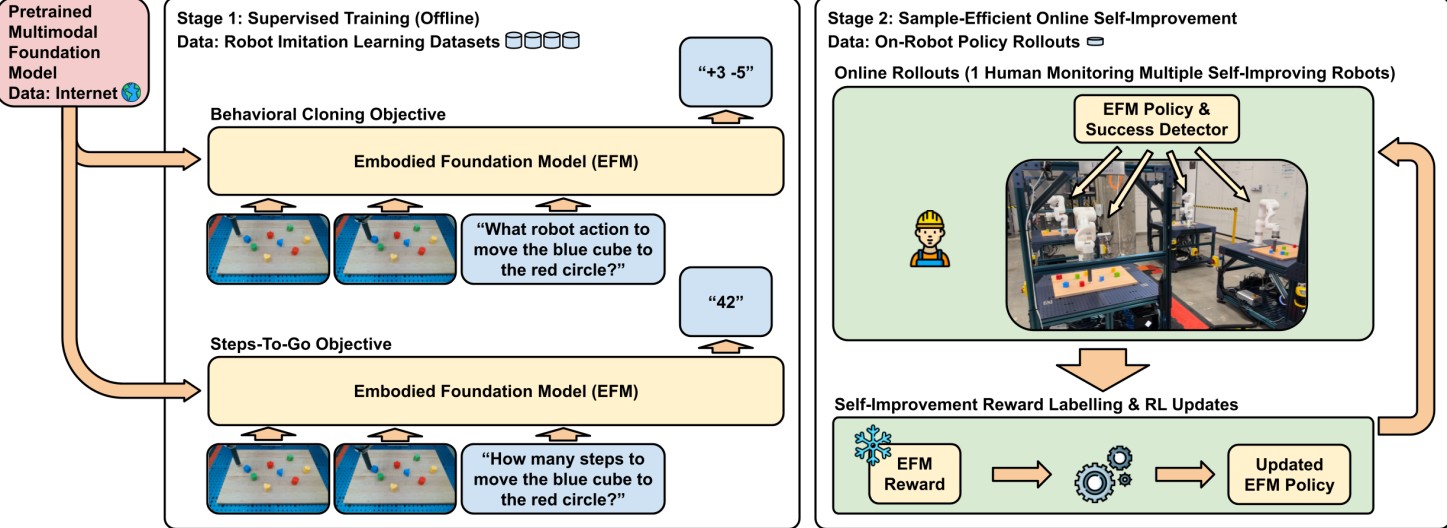

Figure 1: **Overview of our proposed two-stage fine-tuning approach. Stage 1 (Supervised Fine-Tuning):** Starting from a pretrained multimodal foundation model, using robot imitation learning datasets with fine-tune EFMs with a) behavioral cloning, and b) steps-to-go prediction objectives. **Stage 2 (Self-Improvement):** Self-predicted rewards and success detection enable a fleet of robots to autonomously practice downstream tasks with minimal human supervision. Leveraging these self-predicted signals, online reinforcement learning rapidly improves policies and enables the acquisition of novel out-of-distribution tasks.

Despite the unique algorithmic and engineering challenges of investigating RL-tuning of foundation models in the context of real-world robotics, the aforementioned sample-efficiency and performance gains from the LLM literature strongly motivate its investigation. In this work we directly tackle these challenges and design a two-stage framework inspired by LLM post-training processes: In Stage 1 "Supervised Fine-Tuning" (SFT), given an imitation learning dataset we fine-tune EFMs using two objectives: a) behavioral cloning, and b) predicting the number of "steps-to-go" to accomplish desired goals. In Stage 2 "Self-Improvement", we leverage the model's steps-to-go predictions to extract a well-shaped reward function as well as a robust success detector. These key components enable one human operator to monitor multiple robots as they autonomously practice downstream tasks. Critically, our data-driven reward design eliminates the need for ground-truth rewards, and leverages the robustness and generalization properties of the underlying foundation models.

Through extensive experiments on two robot embodiments, LanguageTable [34] and Aloha [57, 3], in the real-world and simulations, we demonstrate the surprising efficacy of our novel post-training framework. First, we demonstrate that not only does Self-Improvement robustly improve policy performance beyond behavioral cloning, but the combination of SFT and Self-Improvement is significantly more sample-efficient than scaling imitation data collection for supervised learning alone. As an example, on the LanguageTable domain [34], 10% additional robot time in the form of Self-Improvement increases policy success rates from 45% → 75%. In contrast, increasing the amount of robot imitation data by 8× leads to a meager 45% → 60% improvement. Further ablations highlight the key role of foundation model pretraining in enabling this sample-efficiency and robustness.

Excitingly, our novel combination of online Self-Improvement and web-scale pretraining also unlocks a unique capability not afforded by prior methods: enabling robots to autonomously practice and acquire new skills. In contrast to prior works that have demonstrated semantic generalization – such as executing the same pick-and-place motions in new contexts [9] – we show that this combination enables behavioral generalization that extends far beyond the imitation data used in Stage 1. Our work highlights the transformative potential of combining pretrained foundation models with online Self-Improvement to unlock autonomous skill acquisition in robotics. Our project website can be found at: `self-improving-efms.github.io`.

# 2 Methodology

Our focus in this work is to investigate the efficacy of RL post-training for embodied foundation models in the context of robotics. However, a critical challenge of reinforcement learning for robotics, and in particular for manipulation tasks, is the problem of reward engineering. Designing effective reward functions requires repeated trial-and-error iterations of training policies and patching reward definitions to mitigate unintended outcomes. Furthermore, even with a perfect definition, measuring rewards in the real-world requires significant engineering effort. Thus, as we move towards a future where we train robots to accomplish increasingly broad sets of tasks, manual reward design becomes untenable for real-world robotics.

We overcome this obstacle via learning data-driven reward functions that also inherit robustness and generalization properties from the web-scale pretraining of the underlying foundation models. Our proposed post-training framework is composed of two stages: 1) Supervised Fine-Tuning (SFT) wherein we fine-tune EFMs using behavioral cloning as well as "steps-to-go" prediction objectives, and 2) Self-Improvement (Online RL) wherein EFMs autonomously practice downstream tasks and rapidly improve via optimizing self-predicted rewards.

## 2.1 Stage 1: Supervised Fine-Tuning (SFT)

The first stage of our framework is the Supervised Fine-Tuning (SFT) stage. Let $\mathcal{D}$ denote an imitation learning dataset. We assume that we can sample tuples $(o_t, a_t, g_{t'}) \sim \mathcal{D}$, where $o_t$ and $a_t$ denote the observation and action at a timestep $t$ respectively, and $g_{t'}$ denotes a goal, event, or outcome that occurs in the future (i.e. $t \leq t'$) within the same trajectory as $(o_t, a_t)$. This assumption subsumes most imitation learning datasets, including hindsight-relabelled as well as single-task datasets[3]. We make no assumptions regarding the optimality of the trajectories in the dataset. Given $\mathcal{D}$, we initialize the EFM using a pre-trained foundation model and perform supervised fine-tuning using the following objectives:

$$\mathcal{L}_{\text{BC}}(\text{EFM}) = -\mathbb{E}_{(o_t, a_t, g_{t'}) \sim \mathcal{D}} \left[ \log p_{\text{action}}^{\text{EFM}}(a_t \mid o_t, g_{t'}) \right]$$

$$\mathcal{L}_{\text{steps-to-go}}(\text{EFM}) = -\mathbb{E}_{(o_t, a_t, g_{t'}) \sim \mathcal{D}} \left[ \log p_{\text{steps-to-go}}^{\text{EFM}}(t' - t \mid o_t, g_{t'}) \right]$$

$\mathcal{L}_{\text{BC}}$ denotes a goal conditioned behavioral cloning loss, where we maximize the log-likelihood of a dataset action conditioned on the observation and the goal. The second objective, $\mathcal{L}_{\text{steps-to-go}}$ teaches the EFM to predict how many timesteps away the policy is from accomplishing the intended goal, given the current observations. This objective plays a critical role in enabling the second post-training stage, Self-Improvement[4].

## 2.2 Stage 2: Self-Improvement

In Stage 2, our goal is to fine-tune the EFM on downstream tasks using online RL in order to rapidly improve policy performance. As we will see later on in our experiments (Sections 4.3.1 and 4.3.2), downstream tasks may even be significantly different than those that appeared in the dataset $\mathcal{D}$ used for Stage 1 training.

**Reward Function**   Let,

$$d(o, g) := \mathbb{E}_{p_{\text{steps-to-go}}^{\text{EFM}}(\text{steps-to-go}|o,g)} \left[ \text{steps-to-go} \right] \tag{1}$$

denote the expected value of "steps-to-go" in order to accomplish goal $g$ given observation $o$, as predicted by the model. The reward function we use for online RL fine-tuning is defined as follows,

$$r(o_t, a_t, o_{t+1}, g) := d(o_t, g) - d(o_{t+1}, g) \tag{2}$$

---

[3]As an example, we can treat a single-task datasets a a goal-conditioned dataset where all episodes share the same goal, and that goal is accomplised at the last timestep of every episode.

[4]In Stage 1 we can include additional auxiliary supervised objectives as well. As an example, in our experiments with the LanguageTable domain, conditioned on the first and last image of an episode we ask the model to predict the instruction that was executed in that episode.

---

**Algorithm 1:** Self-Improvement

---

**Input:** Initialize the policy $p_{\text{action}}^{\text{EFM}}$ from a Stage 1 checkpoint. Initialize and freeze a separate
Stage 1 checkpoint for reward computation and success detection.

**while** *true* **do**
    Initialize empty replay buffer;
    **while** *replay buffer smaller than $N \times B$* **do**
        Sample instruction $g$;
        Execute current policy $p_{\text{action}}^{\text{EFM}}(a_t|o_t, g)$ and end the episode if one of the following
          conditions is met:
            • The success detector indicates success: $\text{success}(o_t, g) == 1$
            • The maximum episode length is reached
            • The human operator manually terminates the episode
        Compute Monte Carlo returns using Equation 2: $R_t \leftarrow \sum_{i=t}^{T} \gamma^{i-t} \cdot r(o_t, a_t, o_{t+1}, g)$;
        Place $(o_t, a_t, g, R_t)$ tuples in the replay buffer;
    **end**
    Perform $N$ policy updates using REINFORCE loss $\left[ -c \cdot R_t \cdot \log p_{\text{action}}^{\text{EFM}}(a_t|o_t, g) \right]$;
**end**

---

Intuitively, this reward function predicts how much closer the robot got towards accomplishing goal $g$ after taking action $a_t$. As the reward function is derived from $d(o, g)$, which is a function of the EFM itself, we refer to our RL fine-tuning process as "Self-Improvement". The choice of using the expected value in Equation 1 is for simplicity and alignment with the notion of a value function in RL (Section G). We leave investigations of alternative definitions such as CVaR [4] for risk-aware policies, or distributional RL [5], to future work.

**Success Detection**    It is important for robot episodes to terminate upon successfully accomplishing the intended goal. Otherwise, a significant portion of the collected data will include the robot resting in a successful state. In settings where we do not have a ground-truth success detector, as in real-world experiments, we use the following success indicator derived from the model,

$$\text{success}(o, g) := \mathbb{1}[d(o, g) \leq s] \tag{3}$$

with $s$ being a very small number of timesteps . We found this formulation of success detection to be very robust even in low data regimes, and significantly more reliable than explicitly including a success detection binary classification objective in Stage 1.

**Self-Improvement**    With the above reward function and success detector in place, we can perform online RL fine-tuning of the EFM on desired downstream tasks. We take a frozen Stage 1 checkpoint for reward function computation and success detection, and initialize the Stage 2 policy from a Stage 1 checkpoint as well [5]. Each iteration of our Self-Improvement loop proceeds as follows: Using the current policy we collect a set of robot trajectories by sampling an instruction $g$, executing the robot policy, and terminating the episode when either 1) the success detector indicates success (i.e. $\text{success}(o, g) == 1$), 2) a pre-specified maximum episode length is reached, or 3) a human operator manually terminates an episode (for example if the robot station gets into a bad configuration). Subsequently, for each timestep in the collected trajectories we compute the Monte Carlo returns $R_t \leftarrow \sum_{i=t}^{T} \gamma^{i-t} \cdot r(o_t, a_t, o_{t+1}, g)$ and place elements $(o_t, a_t, g, R_t)$ in a replay buffer . Once enough data has been collected, we perform $N$ policy updates using the REINFORCE loss,

$$-c \cdot R_t \cdot \log p_{\text{action}}^{\text{EFA}}(a_t|o_t, g) \tag{4}$$

sampling minibatches from the replay buffer without replacement[6]. After $N$ updates, the remaining items in the replay buffer are cleared out and the next iteration begins. Algorithm 1 above presents psuedocode of our proposed Stage 2 Self-Improvement procedure. Although sample-efficiency is

---

[5] Note that these checkpoints are not necessarily identical. For a discussion on checkpoint selection we refer the interested reader to Appendix B.

[6] We use $\gamma = 0.9$, $c = \texttt{5e-2}$. Please refer to Appendix C for further discussion.

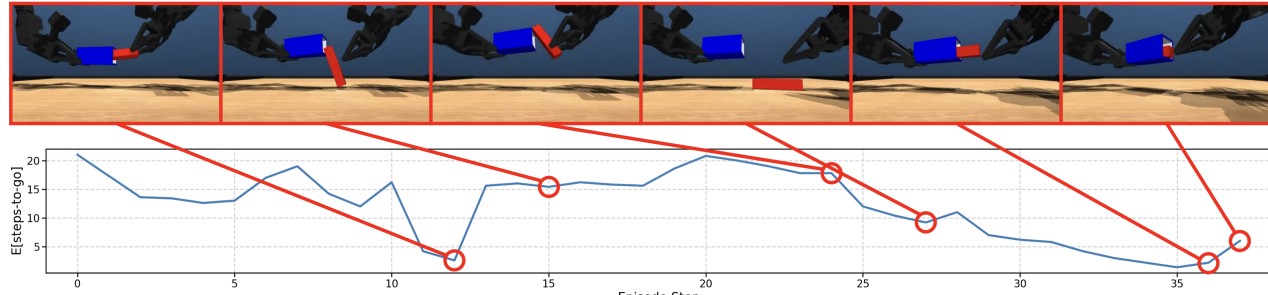

Figure 2: An example trajectory from the Aloha Single Insertion Task and a plot representing $\mathbb{E}[\text{steps-to-go}]$ under the model's prediction (i.e. $d(o, g)$). Key moments: 1) Model believes the episode is about to complete successfully, 2) Policy accidentally drops the peg and $d(o, g)$ increases, 3) Policy regrasps the peg from a bad angle not suitable for insertion so $d(o, g)$ remains high, 4) Policy drops the peg, providing an opportunity to regrasp correctly which reduces $d(o, g)$, 5) Policy is pushing the peg inside and $d(o, g)$ marks that the policy is about to succeed, 6) The right hand knocks the socket out of the left hand's grip which increases $d(o, g)$.

a key consideration of our work, we chose to perform on-policy RL without data reuse. On-policy methods enjoy better training stability, and using REINFORCE specifically obviates the need for training value functions. These choices eliminate two vertices of the deadly triad [51], Off-Policy Learning and Bootstrapping. In Section G we discuss how our choice of reward function leads to a well-shaped objective that reduces the need for baselines in the REINFORCE estimator. We leave the investigation of alternative RL algorithms, including off-policy methods, to future work.

## 3 Intuition on Reward Function

**Visual Intuition**    We can begin to build our intuition regarding the efficacy of steps-to-go prediction by visualizing model predictions on domains of interest. Figures 2 and 8 present visualizations on the Aloha Single Insertion task. In this task, the left arm must first pick up a blue socket, after which the right arm must pick up the red peg and fully insert it into the socket. The captions in these figures walk the reader through the level of intricate details that be can learned during Stage 1 training from the steps-to-go objective. We also encourage readers to visit our supplementary materials website to view additional visualizations such as videos, including on the LanguageTable domain.

Due to space limitations, we continue our discussion in Appendix G. First we discuss the mathematical intuition behind why our algorithmic choices above lead to a well-shaped reward function. Subsequently, we discuss the python notebook in our supplementary material, and demonstrate the efficacy of Self-Improvement on a pedagogical 2D pointmass domain.

## 4 Experiments

In our experiments we seek to validate our proposed Self-Improvement framework and answer the following questions:

- **Q1:** Does Self-Improvement improve performance on downstream tasks beyond the supervised learning stage?
- **Q2:** Is the combination of supervised learning and Self-Improvement more sample-efficient than supervised learning alone?
- **Q3:** Is Self-Improvement, which depends on RL, reliable and reproducible enough to be employed in real-world robotics?
- **Q4:** What is the contribution of pretraining to our Self-Improvement procedure?
- **Q5:** Does web-scale foundation model pretraining enable Self-Improvement on tasks that generalize beyond what was seen in the imitation dataset?

We study these questions using the LanguageTable [34] and Aloha [57, 3] robot embodiments, with experiments in both simulation and the real-world. Throughout this work we use the PaLI 3 billion parameter vision-language model [13, 14] as our base pretrained foundation model. The inputs to our PaLI EFM are images alongside a text sequence representing relevant information such as the instruction, auxiliary information (e.g. joint positions), and whether to predict actions or steps-to-go.

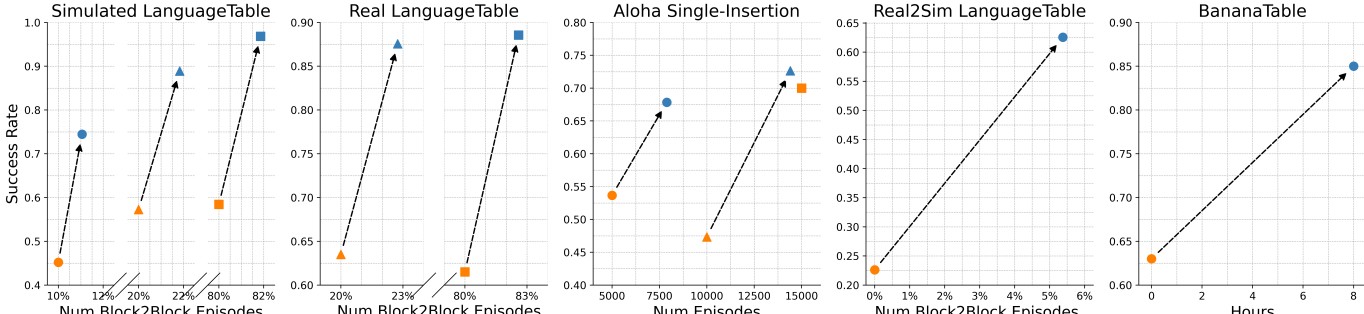

Figure 3: **Stage 2 Self-Improvement Results.** Orange: Stage 1 behavioral cloning policies (equivalent to RT-2 baseline [9]). Blue: Policies after Stage 2 online Self-Improvement with a minimal amount of additional episodes. Results in simulated and real LanguageTable, as well as the Aloha domain, demonstrate that our proposed two-stage post-training approach achieves higher success rates significantly more sample-efficiently than supervised learning alone. Our Real2Sim LanguageTable, and in particular BananaTable results, demonstrate that our novel combination of online Self-Improvement and web-scale pre-training enables policies to rapidly acquire novel skills far outside the Stage 1 imitation learning dataset. Variations across random seeds are small, highlighting the robustness of our approach. Values above are averaged across 3 seeds (unaggregated results in Figures 10 & 11). While Stage 1 LanguageTable datasets contain varied tasks, for fairness the x-axes in the LanguageTable plots above count the number of Block2Block episodes (as a percentage of the total number of Block2Block episodes in the full imitation learning dataset).

The output is a sequence of tokens. To employ PaLI models as policies, we follow the RT-2 [9] policy parameterization and predict tokenized actions. Thus, our Stage 1 behavioral cloning policies are exactly equivalent RT-2 policies which will serve as key baselines. For full details and visualizations regarding models, environments, tokenization, and training we refer readers to Appendix A.

## 4.1 Self-Improvement is Effective, Robust, and More Efficient Than SFT Alone

### 4.1.1 Simulated LanguageTable

The dataset we use to train Stage 1 policies for the simulated LanguageTable domain is the one provided by the original work [34]. This dataset consists of 181,020 human-generated trajectories, with 78,623 unique instructions describing the goals of the trajectories. We subsample this dataset to create 3 new datasets 10%, 20%, and 80% of the original size. For each dataset size, following Stage 1 training we perform Stage 2 fine-tuning with 3 seeds to validate the reliability of our Self-Improvement procedure. We perform Stage 2 fine-tuning on the Block2Block subset of tasks (e.g. `"move the blue moon to the red pentagon"`)[7]. We stop Stage 2 training when policy success rates appear to plateau.

**Results** Figure 3 (first plot) presents our results on the simulated LanguageTable domain, where orange markers represent BC policy performance after Stage 1 (equivalent to RT-2), and blue markers represent policy performance after Stage 2 Self-Improvement. As can be observed, across all dataset sizes (10%, 20%, 80%), our proposed Self-Improvement procedure leads to very significant increase in success rates (minimum 1.5x performance boost), with incredible sample-efficiency in terms of number of episodes (less than 2% extra episodes collected in the Self-Improvement stage). Of particular note, Self-Improvement with 1% additional episodes on top of the 10% dataset size leads to policies that significantly outperform BC policies trained on 20% and 80% dataset sizes. In Appendix K we also show that Self-Improvement is robust and reproducible across random seeds.

### 4.1.2 Real-World LanguageTable

The significant sample-efficiency and robustness of our results above suggest that our Self-Improvement procedure may indeed be applicable for real-world robotics. We apply our proposed approach to the real-world LanguageTable domain, in two settings of using 20% and 80% of the imitation learning dataset [34][8]. As in the simulated setting, we perform Stage 2 fine-tuning on the

---

[7]Our analysis in Appendix L shows that Block2Block instructions make up ∼47% and ∼49% of the instructions in the simulated and real LanguageTable datasets respectively.

[8]We run the 80% data experiment once using 3 robot stations, and run the 20% data experiment twice, once with 3 and once with 4 robot stations.

Block2Block subset of tasks. Given that instruction sampling, reward labeling, and success detection are entirely automated processes, during Self-Improvement *a single human operator is able to monitor our full fleet of LanguageTable robot stations*. The sole responsibility of the human operator is to reset a station if a block falls off the table, or if a station has not been shuffled for 5 minutes. Each experiments is run for approximately 20 hours. For details on the real-world LanguageTable experimentation protocol we refer the interested reader to Appendix F.

**Results** Figure 3 (second plot) presents our results. For both the 20% and 80% data settings, our Stage 2 Self-Improvement procedure improves policy success rate from ∼62-63% up to ∼87%-88%, with only ∼3% additional Block2Block episodes collected. To put this into perspective, this means that with a total amount of experience equivalent to 20% (imitation dataset size) + 3% (Self-Improvement episodes), we obtain policies that far exceed BC (RT-2) policies trained with 80% imitation dataset size! Furthermore, as opposed to the 1-to-1 human-to-robot ratio needed for teleop imitation data collection, Self-Improvement requires only a fraction of the human effort due to the 1-to-many human-to-robot ratio enabled by our proposed approach.

### 4.1.3 Simulated Aloha Single Insertion Task

We also validate our proposed fine-tuning framework on a second robot embodiment, the bimanual Aloha manipulation platform [57, 3]. We design and collect data for a bimanual insertion task, where the left gripper must pick up a socket, and the right gripper must pick up a peg and insert it into the socket. Figure 7 presents a visualization of this task, with videos available on our supplementary materials website. Due to the much more complex observations, 70-dim action space, and much smaller imitation datasets, this presents a challenging setting for further validation of our proposed approach. For details on the environment and data collection process, we refer readers to Appendix D.3. We create 3 imitation dataset sizes of 5K, 10K, and 15K episodes. We apply our two-stage fine-tuning on 5K and 10K dataset sizes, and report results for supervised learning on the 15K dataset as well to better situate the numbers. The differences in methodology compared to experiments in the LanguageTable domain are the following: 1) Checkpoint selection for Stage 2 policy initialization (Appendix B), 2) We noticed that the success condition (the peg fully reaching the end of the socket) is not observable from the cameras, so we add a small positive constant to the reward function when the success condition is met.

**Results** Figure 3 (middle) presents our results. As can be seen, for both dataset sizes Self-Improvement significantly improves policy success rates. As before, we also notice significant sample-efficiency gains where policies trained with 5K (imitation) + 2.5K (Self-Improvement) episodes outperform policies trained with 10K imitation episodes (i.e. RT-2), and rival the success rate of those trained with 15K imitation episodes.

> **A1** Self-Improvement significantly improves policy performance beyond the supervised learning stage.
> **A2** The combination of supervised learning + Self-Improvement is much more sample-efficient than supervised learning alone.
> **A3** Self-Improvement is robust and effective for real-world robot learning.

### 4.2 Importance of Foundation Model Pretraining

It is critical to study to what extent the success of our proposed Self-Improvement procedure is afforded by the web-scale pretraining of the PaLI [13, 14] vision-language foundation model we start from. To ablate the effect of the multimodal knowledge embedded into PaLI, we run our proposed two-stage fine-tuning process starting from alternative variations of the PaLI model:

- **Scratch:** where we use the PaLI architecture but with randomly initialized parameters.
- **Uni-PaLI:** where the PaLI parameters are initialized from a vision model and language model, each pretrained separately, unimodally, without any joint multimodal vision-language fine-tuning. For details please refer to Section A.1.

We compare these variations using an identical setup as Section 4.1.1 on the Simulated LanguageTable domain. Despite our best efforts and very long training runs, we observed that Stage 1 BC policies

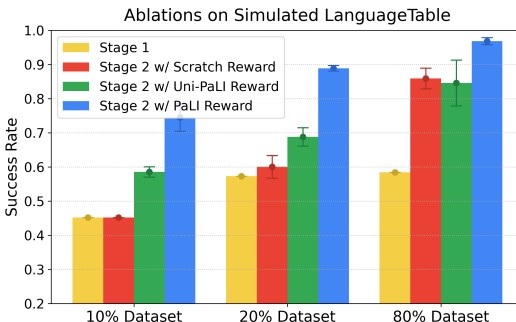
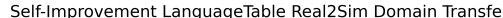
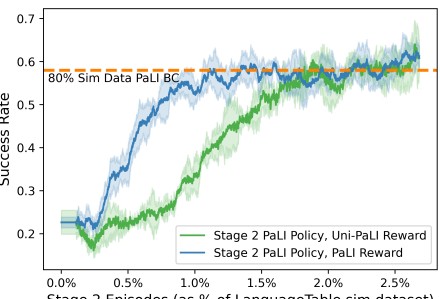

Figure 4: **Left** Ablation results demonstrate the critical role of the web-scale pretraining of foundation models for enabling effective Stage 2 training, in particular in the small dataset size regime. **Right** "Success Rate" plots during Stage 2 Self-Improvement on the LanguageTable Real2Sim domain transfer task. Reward labels from the PaLI model lead to significantly faster Self-Improvement in comparison to the Uni-PaLI model.

derived from the Scratch and Uni-PaLI variations very significantly underperformed PaLI BC policies. Hence, we focus our ablations on the Self-Improvement stage, where we initialize policies from PaLI Stage 1 checkpoints, and use Scratch or Uni-PaLI checkpoints for reward computation.

**Results**    Figure 4 (left) presents our results. There is a clear ordering in performance: PaLI reward models are best, followed by Uni-PaLI, and then Scratch. Scratch reward models lead to high variance results across random seeds, and struggle to provide any meaningful improvements in low-data (10% & 20%) regimes. While better than Scratch, Uni-PaLI reward models perform significantly worse than PaLI reward models across the board, with the gaps more pronounced at lower data settings. In fact, Self-Improvement with the PaLI reward model in the 20% dataset size regime leads to better policies than Self-Improvement with the Uni-PaLI reward model in the 80% regime! These results clearly demonstrate the immense value of multimodal pretraining for Self-Improvement.

> **A4:**    Multimodal pretraining leads to significantly better Self-Improved policies, and is a key enabler of sample-efficiency.

## 4.3    Generalization

The novel combination of our proposed online Self-Improvement process and the use of pretrained multimodal foundation models unlocks a unique capability: enabling policies to practice novel tasks that generalize beyond what was covered by the Stage 1 imitation learning datasets. In this section we present results for two increasingly difficult forms of generalization.

### 4.3.1    Domain Transfer Between Simulation and Real

Starting with a simpler form of generalization, in this section we investigate domain transfer between simulation and real. Sim2Real is an important class of approaches that can significantly reduce the amount of real-world experience needed to train performant robot policies, and has been successfully applied in many settings [43, 48, 2, 44, 30]. To make experimentation simpler, in this section we investigate the inverse problem, Real2Sim transfer, on the LanguageTable domain. We train Stage 1 models using 80% of the *real-world* LanguageTable dataset, and perform Stage 2 Self-Improvement in the *simulated* LanguageTable environment. Similar to our ablation in Section 4.2, we also train Stage 2 models using the Uni-PaLI reward model variant to highlight the role of foundation model pretraining in enabling domain transfer.

**Results**    Figure 4 (right) presents our results. With only 3% extra episodes in the target domain (simulated LanguageTable), our Self-Improvement procedure improves policy performance from ∼22% to ∼59%. This performance is equivalent to BC policies trained with 80% of target domain's imitation dataset. Additionally, Figure 4 (right) demonstrates that the Uni-PaLI reward model leads to a significantly slower Self-Improvement procedure, highlighting the key advantage of pretraining. Given our strong real-world LanguageTable results in section 4.1.2, we expect our Real2Sim results to be strongly indicative of Sim2Real transfer as well.

**Before Stage 2 Fine-Tuning (Instruction: Move the banana to the left center of the board)**

**After Stage 2 Fine-Tuning (Instruction: Move the banana to the top center of the board)**     **LanguageTable Robot View**     **BananaTable Robot View**

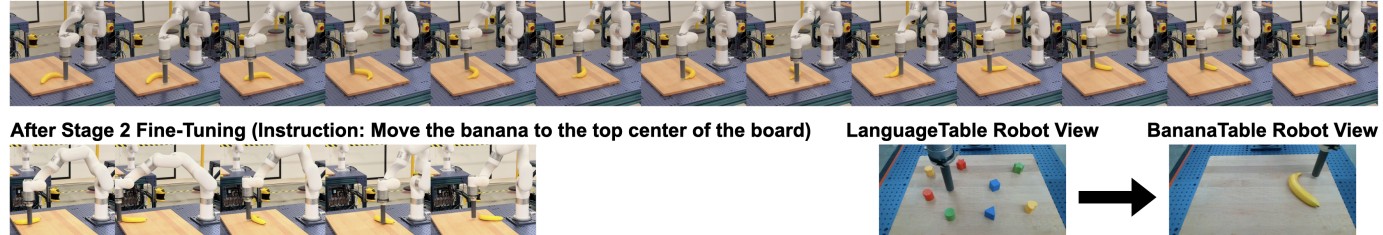

Figure 5: **Strong Generalization to BananaTable. Top** Before Stage 2 fine-tuning on the BananaTable domain, the policy struggles to effectively maneuver a banana across the table due to the difficult geometry. **Bottom Left** After Stage 2 fine-tuning policies are visibly more proficient at the BananaTable task (videos on our supplementary website). **Bottom Right** Prior to Stage 2 BananaTable fine-tuning, the policy and reward models have never seen the BananaTable task, creating a very challenging generalization problem.

### 4.3.2  Strong Generalization to Learning Novel Skills

Moving towards a stronger form of generalization, we investigate whether Self-Improvement with pretrained foundation models enables policies to practice and acquire novel behavioral skills beyond those observed in the imitation learning datasets used in Stage 1. Starting from a policy and reward model trained with the real-world LanguageTable dataset[9], we perform Self-Improvement on a new task we dub "BananaTable" (Figure 5). In this task we replace the LanguageTable blocks with a single prosthetic banana and request policies to push the banana to various locations on the board (e.g. "move the banana to the left center of the board"). The LanguageTable dataset contains no bananas, nor any episodes where the blocks are not on the table. Thus we are solely relying on the generalization abilities of the underlying PaLI foundation model from which the policy, reward function, and success detector are derived.

In contrast to prior works that have demonstrated semantic generalization abilities of robot foundation models (e.g. executing the same pick and place motions in novel contexts in RT-2 [9]), transfer to the BananaTable task requires behavioral generalization, necessitating policies to learn new skills. As an example, due to its elongated geometry, inaccurate pushing of the banana results in it rotating around itself instead of moving in the intended direction (Figure 5, top).

**Results**  Within ~8 hours of Self-Improvement using 2 robot stations, the policy success rate improves from ~63% to ~85%. Beyond the quantitative results, videos on our supplementary website (as well as Figure 5) demonstrate that the policy becomes visibly more proficient at accomplishing the BananaTable tasks, as it picks up on effective strategies for moving the banana around the table. After Self-Improvement, the policy learns to push from either the middle or the tips rather than the rest of the banana to prevent it from rotating around itself.

> **A5:**  The novel combination of our proposed online Self-Improvement procedure and web-scale pretrained foundation models enables policies to rapidly acquire new skills that generalize far beyond the tasks covered by the imitation learning datasets they are provided.

## 5  Related Works

Due to space limitations, here we include a more concise overview of related works. For a more detailed discussion, please refer to Appendix I.

**Embodied Foundation Models**  A number of prior works have leveraged pretrained multimodal foundation models as robot policies, by incorporating action prediction heads. Brohan et al. [9] discretize continuous robot actions and map them onto language model token spaces. Other examples architectures include diffusion models [39, 54] (building on diffusion policies [15, 28]), flow matching [7, 29] (building on [33]), and regression [31]. Critically, these prior works only leverage supervised learning, and the delineation between pretraining and post-training is the task and data

---

[9]We initialize the BananaTable Self-Improvement procedure using the reward model and Self-Improved policy from the 80%-data experiment in Section 4.1.2.

mixture used. To the best of our knowledge, our work is the first to move past supervised learning for training robot foundation models. The key contribution of our work is to present a general-purpose, reward-engineering-free, online Self-Improvement procedure that not only leads to rapid policy improvement, but enables acquisition of novel behaviors outside of what the models have seen in their training data. This form of behavioral generalization is only achievable through online post-training.

**Improving Robot Policies Without Ground-Truth Rewards**   A key obstacle of general-purpose Self-Improvement through RL is that commonly we do not have access to ground-truth reward functions and success metrics, either due to the challenge of designing one (reward engineering), or difficulty in measuring them in the real world (reward instrumentation). A line of prior work, dubbed "Code-As-Rewards", leverages LLMs to write code for reward engineering [36, 56, 53]. Such approaches have a number of downsides that make them impractical for general-purpose robot learning in the real world (Appendix I). Aside from Code-As-Rewards, there exists a rich literature on data-driven reward functions. An important class of works [6, 35, 46, 12] learn latent observation representations on top of which reward and value functions can be defined (e.g. via L2 distance in latent space to target goals). Other approaches include creative relabeling techniques [32, 11, 8] and RL objectives [22, 11]. In comparison, the key advantage of our proposed approach is the straightforward integration with web-scale pretrained foundation models. The closest related works to ours are those based on learning distances in timesteps [26, 27]. Aside from key differences in settings, our focus in this work is on post-training Embodied Foundation Models and generalization, demonstrating that steps-to-go policy improvement is a viable path towards general-purpose Self-Improving robot policies. Additionally, we present steps-to-go thresholding as a novel path towards obtaining robust open-ended success detectors, which have typically been trained via binary classification [19]. Lastly, Yang et al. [55] extend our approach to real-world simulators built on generative video foundation models, and Ma et al. [37] present an extension to an in-context learning formulation.

## 6   Future Work and Limitations

Due to space limitations, here we include a more concise overview of future work and limitations. For a more detailed discussion, please refer to Appendix J. Our work has clearly demonstrated the immense potential of the novel combination of pretrained multimodal foundation models with online Self-Improvement. This combination not only enables very efficient policy improvement on real-world robots, but also unlocks strong generalization capabilities and autonomous acquisition of new skills. There still exist, however, many important avenues for future work. **Algorithmic:** In this work we leveraged on-policy REINFORCE for simplicity which does not reuse any collected data during Self-Improvement. Off-policy methods have the potential to even more substantially boost sample-efficiency. **Skill-Chaining:** How can Self-Improvement and our proposed success detector be extended to enable skill-chaining and solving long-horizon tasks? **Robustness:** Since imitation datasets typically contain expert behavior, they do not contain many forms of failure cases and recovery behaviors. How can our post-training approach be extended to ensure the reliability of reward models and success detectors on out-of-distribution failure cases? We hope that the strong results presented in this work motivate investigation of these fruitful research avenues.

## 7   Conclusion

Drawing inspiration from the success of the reinforcement learning stage in fine-tuning large language models, in this work we proposed a two-stage post-training approach for Embodied Foundation Models. The first stage, Supervised Fine-Tuning (SFT), fine-tunes pretrained multimodal foundation models using two objectives: a) behavioral cloning, and b) steps-to-go prediction. The second stage, online Self-Improvement, leverages steps-to-go prediction for RL post-training. With no task-specific reward engineering and minimal human supervision, this stage enables a fleet of robots to autonomously practice downstream tasks and aquire new skills. Through extensive experiments in the real-world and simulated domains we have demonstrated that Self-Improvement significantly improves policy performance beyond the supervised learning stage, and that the combination of supervised learning and Self-Improvement is much more sample-efficient than supervised learning alone. We then showed that combining web-scale pretraining with Self-Improvement leads to significantly better Self-Improved policies, and is a key enabler of sample-efficiency. Finally, we demonstrated that this novel combination uniquely unlocks a capability not possible by current methods: autonomously aquiring new skills that generalize far beyond the tasks covered in the imitation learning datasets. These findings highlight the transformative potential of combining pretrained foundation models with online Self-Improvement to enable autonomous skill acquisition in robotics.

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

# A Implementation Details

## A.1 Background

**PaLI Vision-Language Foundation Model** Our investigations in this work are independent of the choice of underlying foundation model used. Throughout this work we use the 3 billion parameter PaLI-3B [13, 14] vision-language model as the base pretrained foundation model that we fine-tune for robotics tasks. A PaLI model receives as input one or more images alongside text, and provides text as output. At a high level, the PaLI architecture is comprised of two components: 1) a Vision Transformer (ViT) [42], and 2) an encoder-decoder Transformer [52]. Input images are processed by the ViT into a sequence of "visual tokens". The sequence of visual tokens is concatenated with the tokenized text input and fed into the Transformer which outputs text tokens. The weights of the PaLI model are initialized from a Transformer language model and ViT vision model that are pretrained separately in a unimodal fashion. Following this initialization, the model is fine-tuned with a variety of vision-language training objectives to obtain a multimodal foundation model. For further details regarding the PaLI model, we refer the interested reader to [13, 14].

**RT-2** Brohan et al. [9] introduce a model family, dubbed RT-2, that enables vision-language foundation models (VLMs) to directly produce low-level robot actions for closed-loop control. The two VLMs considered in that work are PaLI [13, 14] and PaLM-E [18], both of which take images alongside text as input, and provide output in the form of text tokens. To enable these VLMs to act as robot policies, continuous robot actions are discretized and mapped onto the text token space of the VLMs. Given image and text inputs, the VLMs are fine-tuned via behavioral cloning (i.e. supervised learning) to predict the tokenized robot actions. While the methods we present in this work are independent of the choice of underlying model, throughout this work our robot policy architectures are equivalent to RT-2 using the PaLI VLM.

As we use the RT-2 policy representation, we also decided to model steps-to-go predictions by discretizing the range of possible number of steps, and mapping them onto the PaLI VLM token space.

## A.2 Environments, Tasks, and Tokenization

For details about the environments and tasks used in this work, please refer to Appendix D.

For details about tokenization, please refer to Appendix E.

## A.3 Training Details

**Stage 1 (Supervised Fine-Tuning)** During the supervised training stage we uniformly distribute each training batch amongst the objectives used in Stage 1. For all domains considered in this work this includes the a) behavioral cloning, and b) steps-to-go prediction objectives. In the real-world and simulated LanguageTable domain experiments, we have an additional objective c) predicting the episode instruction given the first and last frame of an episode. We did not ablate the value of incorporating this objective during training. We used batch size 128 during this stage, used the AdamW optimizer, and trained the entire PaLI model (i.e. kept no component frozen).

**Stage 2 (Self-Improvement)** During this stage we used batch size 64 to require less real-world rollouts for a given number of desired training steps. We kept the ViT portion of the model frozen, intuitively believing that the model has already learned visual features for the task, and that freezing the ViT may potentially help with model stability. We did not ablate this decision. We used the same AdamW optimizer as in Stage 1. The algorithm box in Section 2.2 presents the psuedocode for our proposed Stage 2 Self-Improvement procedure. In each RL loop, we collect enough robot trajectories to perform 16 model update steps ($N = 16$). Intuitively, decreasing $N$ reduces off-policiness of the RL updates, while increasing $N$ improves the diversity of data in the replay buffer due to the larger number of trajectories being collected before performing $N$ RL updates.

## A.4 Compute Resources

Stage 1 (SFT) training was done using one of the following configurations, interchangeably:

- 64 TPUv4 (2x4x4)
- 128 TPUv3

For Stage 2 (Self-Improvement) we used:

- Half of SFT stage resources for the learner job (since we used half batch size)
- 4 TPUv4 (2x2x1) for the reward model
- 4 TPUv4 (2x2x1) for the success detector

## B    Checkpoint Selection for Stage 2 Initialization

The frozen checkpoint used for reward computation and success detection is not necessarily identical to the checkpoint used for policy initialization since the best performance for steps-to-go and behavioral cloning (BC) objectives can happen at different points over the course of Stage 1 training. For the most part throughout this work we took the checkpoints at the best validation loss for the corresponding objective. An exception to this was how we chose the policy initialization checkpoint in the Aloha domain. We observed that at the best validation loss ($\sim$5K-10K steps into training) the BC policy did not have a reasonable success rate. Allowing the model to continue training for much longer and overfitting the validation loss ($\sim$100K-300K steps into training) improved the policy success rate substantially.

## C    REINFORCE Multiplicative Constant

We perform policy updates using the REINFORCE loss,

$$-c \cdot R_t \cdot \log p_{\text{action}}^{\texttt{EFA}}(a_t|o_t, g)$$

In simulation experiments we found that using a small positive multiplicative factor $c$ in the REINFORCE loss plays a significant role in ensuring the model trains stably. Note that this is not equivalent to scaling the learning rate due to interactions with regularizers such as weight decay. Throughout this work we use $c = \texttt{5e-2}$. We did not perform any careful tuning of $c$, and chose its approximate scale using the following intuition: Let $\gamma$ denote the discount factor being used. If we assume the policy gets $N$ steps closer to the goal after every timestep, we have,

$$R_t = \sum_{i=t}^{T} \gamma^{i-t} \cdot N \simeq \frac{N}{1-\gamma}$$

Intuitively, we would like to make the weights on the log probability fall approximately into the range -1 to 1 (i.e. $-c \cdot R_t \in [-1, 1]$). Thus we have $c = \frac{1-\gamma}{N}$. We use $\gamma = \texttt{0.9}$ throughout this work, and hypothetically assume that the range of $N$ is approximately $[-2, 2]$ (e.g. we believe the Stage 2 policy can become twice as efficient as the BC policy)[10]. This results in our choice of $c = \texttt{5e-2}$.

## D    Environments and Tasks

### D.1    LanguageTable

Figure 6 shows the real-world and simulated LanguageTable environments used in this work. The LanguageTable domain [34] has a 2D action space representing delta movement in the x-y plane. The dataset we used in Stage 1 (SFT) are the ones provided by the original work [34] introducing this domain. This dataset consists of 181,020 human-generated trajectories, with 78,623 unique instructions describing the goals of the trajectories. The tasks we perform Stage 2 (Self-Improvement) on are the Block2Block subset of tasks which contain instructions of the form "`move the blue cube to the green star`". As noted in Appendix L, for the simulated and real dataset respectively, $47\%$ and $49\%$ of the instructions fall under the Block2Block tasks. The two images given to PaLI represent the current and previous frame as viewed by the LanguageTable robot camera.

---

[10]Note that this is an approximate intuitive guess and does not need to be precise. A poor guess simply affects the scale of the loss and does not constrain policy learning in any manner.

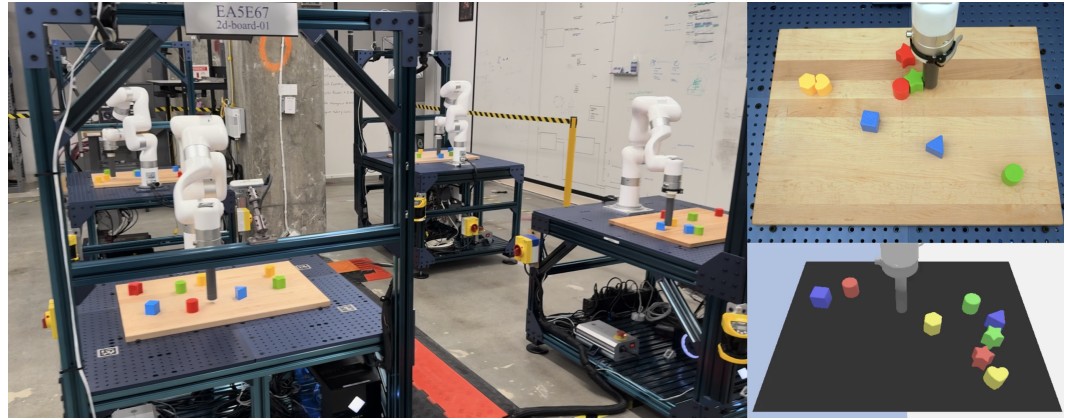

Figure 6: **LanguageTable Environments. Left** The four LanguageTable robot stations used for our real-world experiments. **Right, Top** Camera view of the real-world LanguageTable robot station. **Right, Bottom** Camera view of the simulated LanguageTable robot station.

## D.2 BananaTable

In the BananaTable task we remove all blocks from the LanguageTable stations and replace them with a single banana. The instructions for the BananaTable task have the form, "`X the banana to the Y of the table.`", where `X` is a set of verbs synonomous with pushing, and `Y` is one of `left, top left, top center, top right, right, bottom right, bottom, bottom left, center.`

## D.3 Aloha

The Aloha domain [57] is a bimanual robot station with 14 degrees of freedom and controlled via joint position commands. As opposed to the default of predicting 50Hz actions, we predict 10Hz actions. A common design choice in the Aloha domain is to train policies to predict $N$ actions into the future. This is commonly referred to as action-chunking [57], or action horizon [15]. We use $N = 5$ which results in an action space that is 70-dimensional ($14 \times 5$). During rollouts, we execute the full action-chunk. In the Aloha domain, as input we also provide the model with the current joint positions, i.e. we append 14 tokens to the input text instructions, where each token represents a number from 0-255. For details on tokenization, please refer to Appendix E.

The Aloha environment has 4 cameras (Figure 7, right). To turn them into two images to pass to our PaLI models, we vertically stack two images into one image with a black buffer in between. We stack the top and table view images to form the first image, and stack the left and right wrist view to form the second image. We add a small black band between the stacked views inside each image in hopes of better delineating them. Since we pass 224×224 images to PaLI, this means that each Aloha camera view appears with an effective resolution of about 100×100. This is significantly less resolution than the typical Aloha resolution of 480×640 [58].

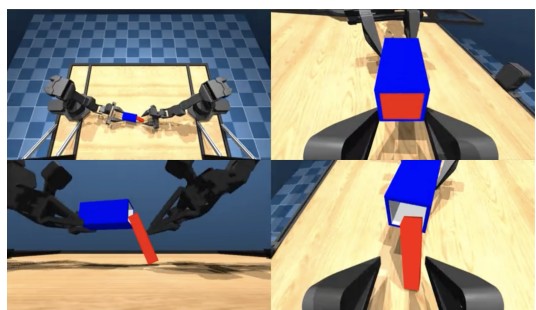

Figure 7: The four camera views in the simulated Aloha Single-Insertion task.

We designed and collected data for a bimanual insertion task, where the left gripper must pick up a socket, and the right gripper must pick up a peg and insert that peg into the socket. We collected 800 demonstrations using a VR headset to display the Mujoco simulation, and using the real-world Aloha leader robots to control the virtual robots. We then trained a small diffusion policy [15] on the 800 demonstrations and used the model to generate 3 datasets of size, 5K, 10K, and 15K. Note that these datasets only contain successful rollouts, and max episode lengths was chosen generously (1500 steps) to allow for recovery from mistakes.

*Critical to successful PaLI policies was to employ semi-global action representations as in Chi et al. [16]*, as well as training Stage 1 (SFT) far beyond the point at which the best validation loss was obtained for the behavioral cloning loss (Appendix B).

# E    Tokenization

## E.1    Real/Sim LanguageTable & BananaTable

We use the same tokenization approach for the real-world and simulated LanguageTable, as well as the BananaTable domains.

**Action Tokenization**    As noted in Appendix D.1, the above domains have a 2D continuous action space. We represent LanguageTable actions via a sequence of 4 tokens:

1. token for $+/-$
2. token representing a number in the range $[0, 10]$
3. token for $+/-$
4. token representing a number in the range $[0, 10]$

The continuous 2D actions are binned to fall into this representation.

**Steps-to-go Tokenization**    We computed the upper percentile of episode lengths in the imitation dataset to be 100 steps. We discretized the range from 0 to 100 steps into 50 bins, and represented each bin using a single token.

## E.2    Aloha

**Action Tokenization**    As discussed in D.3 our action space is $5 \times 14$ dimensions. We represent each dimension with 1 token, meaning the model outputs 70 tokens. Each token represents a number from 0-255. The continuous Aloha actions which are in the range $[-1, 1]$ are discretized and binned into these 256 bins.

**Steps-to-go Tokenization**    The upper bound on episode lengths in the imitation dataset was 1500 steps. Since we train policies with action-chunk 5 [57] and execute the full action-chunk during rollouts, this reduces the maximum episode length to 300 steps. We represent the range from 0 to 300 using a single token per number.

**Joints Tokenization**    In the Aloha domain, as input we also provide the model with the current joint positions. We append 14 tokens to the input text instructions, where each token represents a number from 0-255. The continuous Aloha joints which are in the range $[-1, 1]$ are discretized and binned into these 256 bins, in an identical manner as the actions.

# F    Real-World LanguageTable Experimentation Procedure

For all real-world experiments, 1 human was responsible for monitoring all robots and performing resets. They did not provide any form of labels or success indicators to the models. Operators were instructed to perform resets either when a block drops off the table, or if a station has not been shuffled and reset in the past 3-5 minutes of operation.

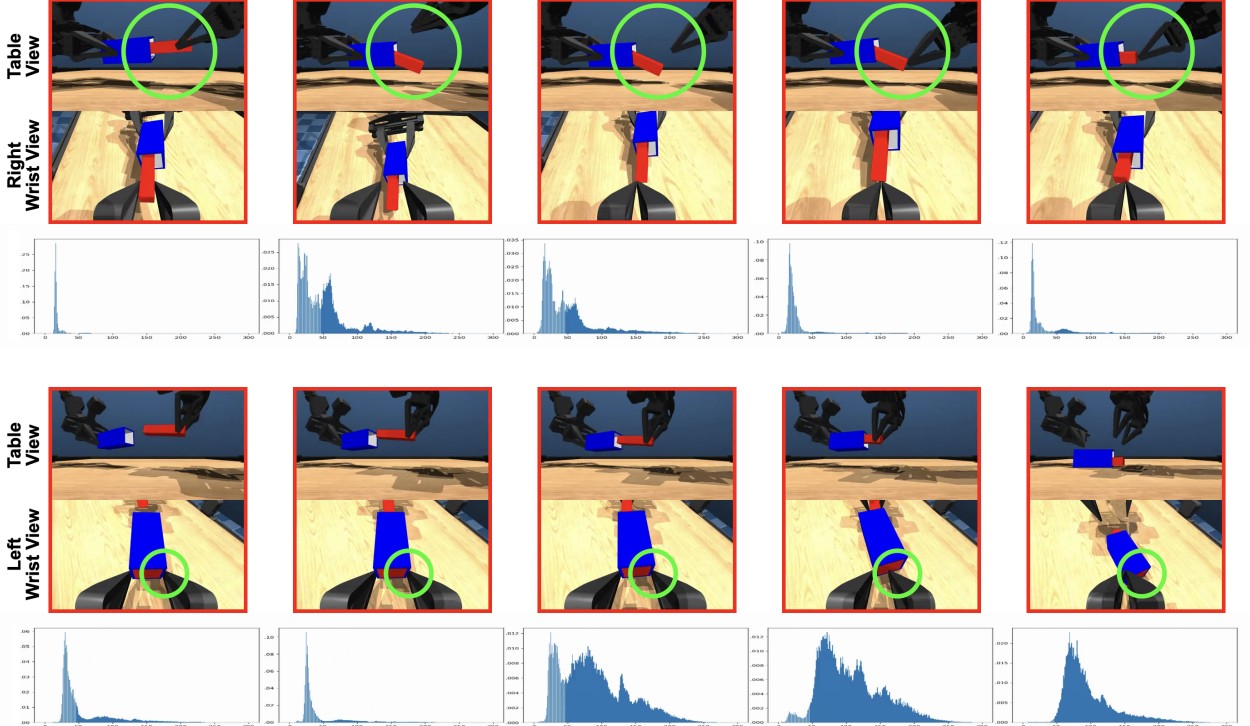

Figure 8: The two figures above demonstrate the intricate level of detail that a model learns from the steps-to-go prediction objective in Stage 1. Each figure captures an interesting moment in an Aloha Single Insertion task rollout. Each is comprised of 5 consecutive frames, where below each frame we visualize the probability distribution of the model's prediction for steps-to-go until success. The x-axis represents the number of steps-to-go, and the y-axis represents the probability mass. **Top** In the first frame, the policy is about to successfully insert the peg and complete the task, so the model predicts that with high likelihood the policy will succeed soon. However, in the next frame the policy lets go of the peg too soon and the peg is about to fall. Thus the predicted steps-to-go widens drastically into a multimodal distribution, considering the spectrum of possibilities from a quick recovery to longer recovery times. As the policy recovers in the fourth and fifth frames, the model's prediction narrows back to a unimodal distribution, with high likelihood of success in the near horizon. **Bottom** In the first two frames the policy is on track to successfully complete the task, so the model predicts that with high likelihood the policy will succeed soon. However, in the third frame the socket begins to slip out of the left gripper. Despite this slippage being barely visible from the left wrist camera, and not visible in any of the other camera views, the model immediately picks up on this event and its predictions widen significantly with multiple modes. Specifically, the model places some probability mass on an immediate save, and distributes the rest of the probability mass over a range of possible recovery times. In the fourth and fifth frames the socket fully slips out of the gripper, so the model removes the probability mass on the immediate save outcome.

## G    Intuition on Reward Function (Continued)

**Visual Intuition (Continued)**    We invite readers to study Figure 8, which demonstrates the intricate level of detail that a model learns from the steps-to-go prediction objective in Stage 1.

**Mathematical Intuition**    Let $\mu$ denote the policy corresponding to the imitation learning dataset $\mathcal{D}$ (e.g. if the dataset was collected via tele-operation, $\mu$ would represent the "human policy"). Consider the reward function $-\mathbb{1}\Big[o_t \text{ satisfies } g\Big]$ that is 0 when the goal is satisfied, and -1 elsewhere. Let $V^\mu$ denote the undiscounted value function of policy $\mu$ for this reward function. We have,

$$V^\mu(o_t, g) = \mathbb{E}_\mu\left[\sum_{i=t}^{T} -\mathbb{1}\Big[o_i \text{ satisfies } g\Big]\right] = \mathbb{E}_\mu\Big[-1 \cdot \text{steps-to-go}\Big] =: -d(o_t, g)$$

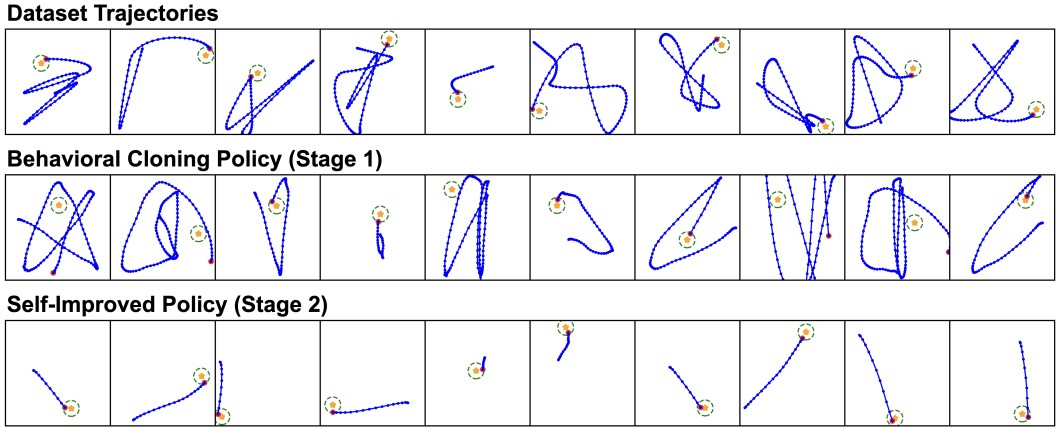

**Figure 9: Pointmass Navigation Domain.** Sample trajectories from the imitation learning dataset, as well as BC (Stage 1) and Self-Improved (Stage 2) policies.

Substituting $V^\mu$ in Equation 2 we obtain,

$$r(o_t, a_t, o_{t+1}, g) = V^\mu(o_{t+1}, g) - V^\mu(o_t, g) = \underbrace{(1-\gamma) \cdot V^\mu(o_{t+1}, g)}_{\text{core reward}} + \underbrace{\left[\gamma \cdot V^\mu(o_{t+1}, g) - V^\mu(o_t, g)\right]}_{\text{reward shaping}}$$

(5)

where $\gamma$ is the discount factor used in the Stage 2 RL updates. We see that $r(o_t, a_t, o_{t+1}, g)$ is implicitly a shaped reward function [38] that provides higher rewards in states where $\mu$ knows how to perform well (i.e. core reward $(1-\gamma) \cdot V^\mu(o_{t+1}, g)$ is high). Thus, *Self-Improvement leads to policies that achieve intended goals more efficiently than the dataset policy $\mu$, while being implicitly regularized to stay close to regions of the state space where $\mu$ is proficient!*

Using Equation 5 to simplify the telescoping sum in the Monte Carlo returns we have,

$$R_t = \sum_{i=t}^{T} \gamma^{i-t} \cdot r(o_i, a_i, o_{i+1}, g) = \left[(1-\gamma) \cdot \sum_{i=t}^{T} \gamma^{i-t} \cdot V^\mu(o_{i+1}, g)\right] - \underbrace{V^\mu(o_t, g)}_{\text{baseline}}$$

The baseline $V^\mu(o_t, g)$ leads to lower variance estimates that are particularly useful in our case of using the REINFORCE estimator. When $\gamma$ is close to 0, we have $R_t = V^\mu(o_{t+1}, g) - V^\mu(o_t, g)$ which is closely similar to a single-step policy improvement for the $-\mathbb{1}[o_t \text{ satisfies } g]$ reward. As $\gamma \to 1$, $R_t$ encourages policies to traverse trajectories along which the states have high value under the dataset policy $\mu$ (i.e. high $V^\mu$).

**Pointmass Navigation Domain**  In our supplementary materials website we include a self-contained python notebook implementing Self-Improvement on a pointmass navigation domain. In each episode the pointmass starts in a random position, and the goal is for the pointmass to reach a different randomly sampled target position. We create a purposely sub-optimal imitation learning dataset for this task, where using a PD-controller we navigate to 5 waypoints before heading to the goal position. We then execute our proposed fine-tuning procedure on this imitation dataset using MLP policy and steps-to-go prediction models. Figure 9 shows sample trajectories from the dataset, as well as BC (Stage 1) and Self-Improved (Stage 2) policies. As anticipated, BC policies mimic the sub-optimalities of the dataset. However, in the second stage, and without access to ground-truth rewards, our proposed Self-Improvement procedure very rapidly brings policies close to optimality. For reproduction using our self-contained Colab notebook, as well as videos visualizing trajectories and steps-to-go predictions, please refer our supplementary materials website.

## H  Future Work and Limitations

**Episode Boundaries & Skill–Chaining**  The steps-to-go auxiliary loss that underpins our Self-Improvement stage naturally lends itself to hierarchical control. By explicitly annotating the start

and termination of sub-skills, the same progress estimator can be reused to produce dense sub-task rewards, enabling long-horizon skill-chaining reminiscent of option-based RL. At inference time, a high-level planner — whether a finite-state machine, a human tele-operator, or a reasoning foundation model — can invoke the learned sub-policies and rely on the steps-to-go predictions to decide when to transition to the next skill. The chief obstacle is scalable episode and skill boundary annotation: manual labeling is prohibitively expensive, calling for creative strategies — such as those leveraging existing multi-modal foundation models — that recover consistent boundaries from raw interaction logs. Exploring such automated segmentation is an exciting avenue for future research.

**Reward Models**   Because reward inference does not have real-time requirements in our framework, latency constraints are minimal; we can therefore allocate far larger models — or even iterative, chain-of-thought style reasoning [25] — to obtain higher-fidelity labels. A key challenge of learning reward models from imitation-based datasets is handling settings where failure states fall outside the support of the datasets, and the absence of recovery trajectories for those out-of-distribution (OOD) states. More expressive steps-to-go estimators leveraging broader data sources — robotics or otherwise — could recognise these OOD states and either assign appropriate shaped rewards or trigger a switch to a recovery skill, thereby improving robustness at deployment. Another potentially promising avenue to handle OOD states is to collect a dataset of robot rollouts using robot policies, and labeling those trajectories for training the steps-to-go estimator.

**Embodied Foundation Models**   Our study fine-tunes general-purpose vision-language backbones that were never exposed to robotics data during pretraining. As larger multimodal corpora of robot experience become available, it will be crucial to design pretraining curricula that endow Embodied Foundation Models with strong priors for physical reasoning, while preserving their broad visual-semantic knowledge [29, 49]. On the post-training side, as opposed to post-training for specific downstream tasks as done in our work, a general-purpose post-training stage analogous to language models could render the resulting policies effective in a purely zero-shot manner, reducing or even eliminating the need for task-specific downstream fine-tuning. As an encouraging sign, preliminary evidence from our LanguageTable experiments hints that strengthening one task (Block2Block) via Self-Improvement can noticeably boost success rates for other instructions (e.g. moving a block to specific locations).

**RL Algorithms**   For simplicity and stability we elected to use on-policy REINFORCE with no data reuse. This choice eliminates two vertices of the deadly triad [51], Bootstrapping and Off-Policy Learning. However, this choice forgoes the data-reuse benefits of modern off-policy algorithms. Investigating off-policy variants that scale to large models [23] stands to further curb robot-hour requirements. Theoretical investigations into our choice of reward function and policy update procedure are also promising avenues for future work. As discussed in Section G, our Self-Improvement algorithm implicitly regularizes policies to remain close to the behvaior cloned policy, but with a distincly different mechanism than the often-used KL regularization approach [25]. Our approach is more broadly applicable to any value function, not just steps-to-go. Lastly, we observe that pushing Self-Improvement beyond its performance peak can degrade success rates, suggesting that better stopping criteria or adaptive regularisers are required to prevent over-optimisation of the shaped reward. Theoretical investigations into our choice of reward function may also uncover the causes of such degradation.

**Summary**   Our proposed Self-Improvement recipe already unlocks substantial gains, with minimal reward engineering and a single human supervising multiple robots. However, addressing the challenges above is essential for scaling towards general-purpose autonomous skill acquisition. We look forward to future research efforts that extend our groundwork to long-horizon tasks, richer reward inference, domain-aligned pretraining, and more sample-efficient reinforcement learning.

# I   Related Works

**Embodied Foundation Models**   A number of prior works have leveraged pretrained multimodal foundation models as robot policies. Driess et al. [18] demonstrate how separately pretrained vision and language foundation models can be co-trained to create multimodal foundation models. They highlight how these multimodal models can learn to query low-level robot controllers towards

accomplishing high-level objectives. Building on this direction Brohan et al. [9] discretize continuous robot actions and map them onto language model token spaces. This method, dubbed RT-2, enables pretrained vision-language foundation models (VLMs) to be fine-tuned as robot policies. This approach was further validated by applying to the Open X Embodiment [17] dataset containing over 1M robot trajectories from 21 institutions, and is the policy architecture we base our work off of. Since RT-2, a variety of works have extended pretrained VLMs by incorporating action prediction heads. Example action head architectures include diffusion models [39, 54] (building on Chi et al. [15], Ho et al. [28]), flow matching [7, 29] (building on Lipman et al. [33]), and L1 regression [31]. Critically, in prior works the delineation between pretraining and post-training is the task and data mixture used to perform offline supervised fine-tuning. To the best of our knowledge, our work is the first to move past supervised learning for training robot foundation models. The key contribution of our work is to present a general-purpose, reward-engineering-free, online Self-Improvement procedure that not only leads to rapid policy improvement, but enables acquisition of novel behaviors outside of what the models have seen in their training data. This form of behavioral generalization is only possible through an online post-training mechanism.

**Improving Robot Policies Without Ground-Truth Rewards** Our goal is to design methods that enable generalist robot foundation models to autonomously become proficient on any downstream task. A key obstacle of general-purpose Self-Improvement through reinforcement learning is that commonly we do not have access to ground-truth reward functions and success metrics, either due to the challenge of designing one (reward engineering), or difficulty in measuring them in the real world (reward instrumentation). A line of prior work, that we dub "Code-As-Rewards", leverages LLMs to write code for reward engineering [36, 56, 53]. Policies are then trained with the designed reward, and the success rates and other feedback are provided to the LLMs for improving the reward function. Such approaches have a number of downsides that make them impractical for general-purpose robot learning in the real world: 1) It is notoriously difficult to arrive at intended policies through reward engineering, in particular for dexterous manipulation tasks, 2) The iteration loop of training a policy with a given reward and patching the reward function based on outcomes is impractical for real-world robotics, 3) The variables needed in such reward functions require bespoke instrumentation to be measurable outside of simulation domains, 4) We still require a ground-truth success detector to provide feedback to the LLM designing the rewards. Aside from Code-As-Rewards, there exists a rich literature on obtaining data-driven reward functions. Such approaches forego manual reward engineering and instead design expressive representations that scale with increasing data. An important class of works [6, 35, 46, 12] learn latent observation representations on top of which rewards and value functions can be defined (e.g. via L2 distance in latent space to target goals). Kumar et al. [32] use a heuristic of labeling imitation learning datasets with $+1$ rewards near successful states, and 0 elsewhere. They demonstrate that offline and online RL, using the combination of target task and pre-existing data, can be used to sample-efficiently improve robot policy performance. Eysenbach et al. [22] design a contrastive learning objective that corresponds to a form of goal-conditioned Q-Learning. Chebotar et al. [11] demonstrate that offline goal-conditioned RL with relabeled goals can lead to sample-efficient downstream fine-tuning for new tasks. In comparison to the above, the key advantage of our proposed approach is the straightforward integration with web-scale pretrained foundation models. RoboCat [8] trains a large behavioral cloning (BC) Transformer with a similar architecture as Gato [45] on a diverse set of robotics tasks. They demonstrate that policy performance can be improved by rolling out BC policies, hindsight relabeling episodes with accomplished goals, and adding the trajectories back into the imitation dataset. However, it is important to note that using hindsight relabeled supervised learning as a policy improvement procedure can have important failure cases [24]. The closest related works to ours are those based on learning distances in terms of timesteps. Hartikainen et al. [26] present an iterative procedure where 1) policies are rolled out to collect trajectories, 2) steps-to-go prediction between states is updated through supervised learning, and 3) the negative distance reward function is used for updating policies through RL. In an offline setting, Hejna et al. [27] also learn steps-to-go between states using supervised learning, estimate shortest paths between states and goals, and use weighted behavioral cloning to obtain improved policies. Aside from important differences in settings, our focus in this work is on foundation models and generalization, demonstrating that steps-to-go policy improvement is a viable path towards general-purpose Self-Improving robot policies. In addition to providing a dense reward signal, our work presents steps-to-go thresholding as a novel path towards obtaining robust open-ended success detectors, which have typically been trained via binary classification [19]. Lastly, we highlight two extensions of our work. Yang et al. [55] demonstrate that

our approach is effective in real-world simulators built on generative video foundation models. Ma et al. [37] present an extension our work. They demonstrate that the long context capabilities of state-of-the-art foundation models enables in-context steps-to-go prediction, which can be used for offline RL, success detection, and dataset filtering, without necessitating foundation model fine-tuning.

## J Future Work and Limitations

**Episode Boundaries & Skill–Chaining**   The steps-to-go auxiliary loss that underpins our Self-Improvement stage naturally lends itself to hierarchical control. By explicitly annotating the start and termination of sub-skills, the same progress estimator can be reused to produce dense sub-task rewards, enabling long-horizon skill-chaining reminiscent of option-based RL. At inference time, a high-level planner — whether a finite-state machine, a human tele-operator, or a reasoning foundation model — can invoke the learned sub-policies and rely on the steps-to-go predictions to decide when to transition to the next skill. The chief obstacle is scalable episode and skill boundary annotation: manual labeling is prohibitively expensive, calling for creative strategies — such as those leveraging existing multi-modal foundation models — that recover consistent boundaries from raw interaction logs. Exploring such automated segmentation is an exciting avenue for future research.

**Reward Models**   Because reward inference does not have real-time requirements in our framework, latency constraints are minimal; we can therefore allocate far larger models — or even iterative, chain-of-thought style reasoning [25] — to obtain higher-fidelity labels. A key challenge of learning reward models from imitation-based datasets is handling settings where failure states fall outside the support of the datasets, and the absence of recovery trajectories for those out-of-distribution (OOD) states. More expressive steps-to-go estimators leveraging broader data sources — robotics or otherwise — could recognise these OOD states and either assign appropriate shaped rewards or trigger a switch to a recovery skill, thereby improving robustness at deployment. Another potentially promising avenue to handle OOD states is to collect a dataset of robot rollouts using robot policies, and labeling those trajectories for training the steps-to-go estimator.

**Embodied Foundation Models**   Our study fine-tunes general-purpose vision-language backbones that were never exposed to robotics data during pretraining. As larger multimodal corpora of robot experience become available, it will be crucial to design pretraining curricula that endow Embodied Foundation Models with strong priors for physical reasoning, while preserving their broad visual-semantic knowledge [29, 49]. On the post-training side, as opposed to post-training for specific downstream tasks as done in our work, a general-purpose post-training stage analogous to language models could render the resulting policies effective in a purely zero-shot manner, reducing or even eliminating the need for task-specific downstream fine-tuning. As an encouraging sign, preliminary evidence from our LanguageTable experiments hints that strengthening one task (Block2Block) via Self-Improvement can noticeably boost success rates for other instructions (e.g. moving a block to specific locations).

**RL Algorithms**   For simplicity and stability we elected to use on-policy REINFORCE with no data reuse. This choice eliminates two vertices of the deadly triad [51], Bootstrapping and Off-Policy Learning. However, this choice forgoes the data-reuse benefits of modern off-policy algorithms. Investigating off-policy variants that scale to large models [23] stands to further curb robot-hour requirements. Theoretical investigations into our choice of reward function and policy update procedure are also promising avenues for future work. As discussed in Section G, our Self-Improvement algorithm implicitly regularizes policies to remain close to the behvaior cloned policy, but with a distincly different mechanism than the often-used KL regularization approach [25]. Our approach is more broadly applicable to any value function, not just steps-to-go. Lastly, we observe that pushing Self-Improvement beyond its performance peak can degrade success rates, suggesting that better stopping criteria or adaptive regularisers are required to prevent over-optimisation of the shaped reward. Theoretical investigations into our choice of reward function may also uncover the causes of such degradation.

**Summary**   Our proposed Self-Improvement recipe already unlocks substantial gains, with minimal reward engineering and a single human supervising multiple robots. However, addressing the challenges above is essential for scaling towards general-purpose autonomous skill acquisition. We

look forward to future research efforts that extend our groundwork to long-horizon tasks, richer reward inference, domain-aligned pretraining, and more sample-efficient reinforcement learning.

# K    Additional Plots

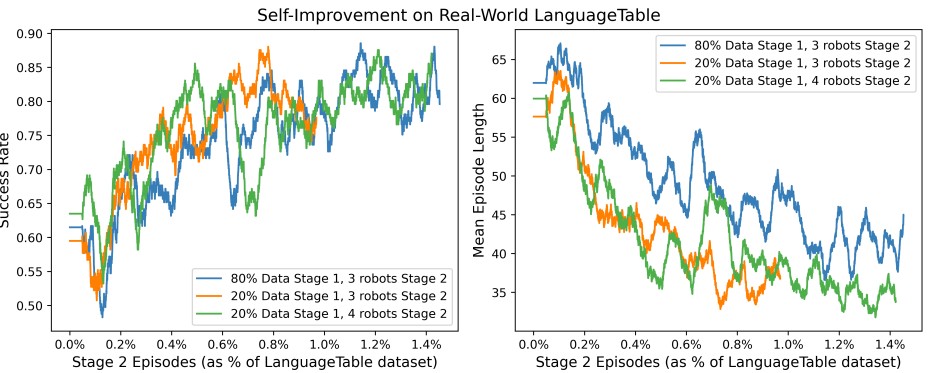

Figure 10: Self-Improvement "Success Rate" plots during Stage 2 Self-Improvement on the real-world LanguageTable domain. We conducted real-world experiment 3 times: 1) 80% imitation dataset size in Stage 1, Stage 2 fine-tuned on 3 robots simultaneously, 2) 20% data in Stage 1, Stage 2 with 3 robots, 3) 20% data in Stage 1, Stage 2 with 4 robots. In all Stage 2 experiments a single human monitored and performed periodic resets for all robots. Each experiment took approximately 20 hours (4 hours × 5 days). The x-axis in the plots aboves shows the amount of extra episodes collected during the Stage 2 online Self-Improvement process, as a percentage of the total number of Block2Block episodes in the LanugageTable dataset.

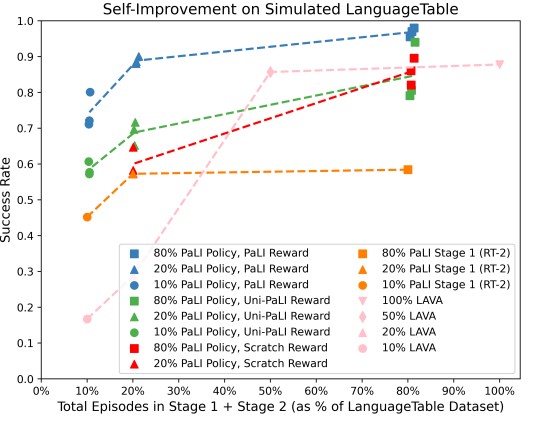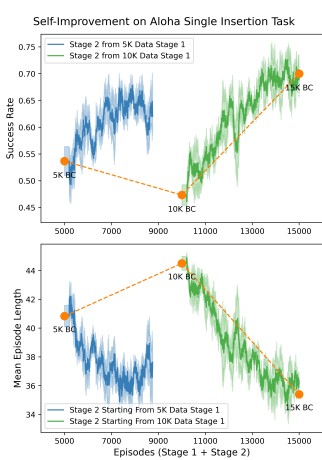

Figure 11: **Left** Results and ablations on the simulated LanguageTable domain. We emphasize to the reader that while it appears that the Stage 1 and Stage 2 plots have identical x-axis values, there is no bug in the plot and they are in fact different. The Stage 2 Self-Improvement process is simply so sample-efficient that the difference in x-axis is not observable in this plot. The first plot in Figure 3 in the main text presents a different view that makes the difference more apparent. **Right** "Success Rate" and "Mean Episode Length" plots during Stage 2 Self-Improvement on the Aloha Single Insertion Task (5K and 10K data settings, 3 random seeds each setting). Similar to the discussion in Section 4.1, the above plots demonstrate that the combination of SFT + Self-Improvement is more sample-efficient than allocating the full robot time budget to collecting imitation data for SFT.

# L  Computing Percentage of Block2Block Instruction in LanguageTable

To get a sense of the percentage of the LanguageTable datasets corresponding to Block2Block tasks, we used Gemini 1.5 Pro and its structured outputs feature to label LanguageTable instructions as being Block2Block or not. For both the simulation and real datasets we randomly sampled $N = 5000$ of the instructions in the dataset, and used the following prompt to classify them as Block2Block. Using the structured outputs feature of Gemini enables us to enforce the LLM responses to be either `Yes` or `No`.

> You are a language model with expertise in determining the structure and type of robotic instructions. Your task is to identify whether a given instruction is a "block to block" type instruction or not. A "block to block" instruction involves moving or pushing one block towards another specific block on a board. This does not include separating two blocks, putting one block in between two blocks, or putting a block near a group of blocks.
> Please analyze the following instruction and respond with either "yes" or "no" based on whether it fits the definition of a "block to block" instruction:
> Examples where the answer is "yes":
> - "push the red circle towards the blue triangle"
> - "push blue cube to the right of green cube"
> - "move the red moon towards the bottom left side of the red pentagon"
> - "push red pentagon into the green cube"
> - "place the yellow star to the left of the red moon"
> - "push the green cube vertically below the yellow pentagon"
> - "drag green star into blue cube"
> - "slide the red star at the bottom right of the green star"
> Examples where the answer is "no":
> - "push the blue cube in between yellow star and green star"
> - "push the red crescent away from the blue crescent"
> - "place the arm to the left of the yellow star"
> - "move the blue crescent to the center of the board"
> - "adjust the group of blocks to form a circle"
> - "separate the green star and the blue cube"
> - "push blue moon along with yellow star to the left side"
> - "move yellow star and blue moon together slightly to the top side of the board"
> - "place the blue cube at the top center"
> - "slide blue cube a bit right"
> The instruction for you to label as "yes" or "no" is:

For the simulated and real dataset respectively, $47\%$ and $49\%$ of the instructions were labeled as Block2Block. Using Hoeffding's Inequality we can see that with $N = 5000$, these estimates are within $2.8\%$ error margin with $99.9\%$ confidence.

# M  Interesting Observations & Incomplete Experiments

In this section we note interesting observations, and highlight results of experiments that we were unable to complete due a change of institution affiliation of the lead author.

**Positive Transfer of Self-Improvement**  In Section 4.1 we note that we perform Self-Improvement on the Block2Block subset of the LanguageTable tasks. We have observed through real-world

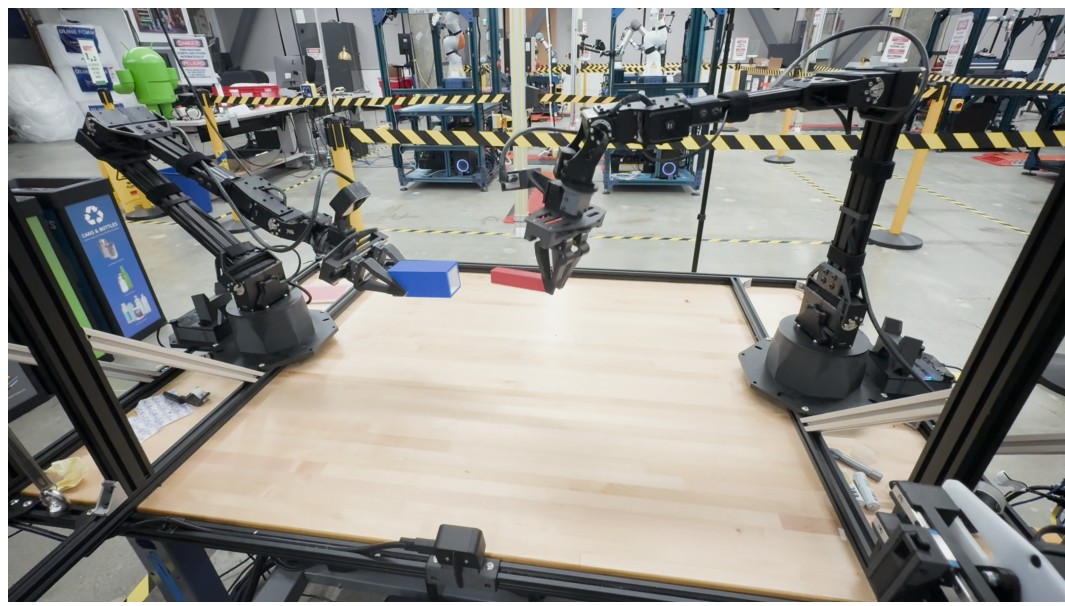

Figure 12: Real-world replica of our simulated Aloha Single-Insertion task

rollouts that Self-Improvement not only significantly improves performance on Block2Block tasks, but actually improves the model across all LanguageTable tasks.

**All-Instructions Experiment in the Real-World LanguageTable Domain**    In the real-world LanguageTable domain we conducted an experiment where after Self-Improvement on Block2Block tasks, we ran Self-Improvement on all possible instructions. Prior to conducting this experiment, we trained a separate PaLI model to produce a goal instruction given the current image. We then started the all-instructions Self-Improvemnet by mixing instructions at a 50-50 ratio between Block2Block and model-generated instructions. Over time we decayed the rate of Block2Block instructions down to zero. We noticed a clear improvement in success rate metrics for all tasks. However, we noticed that for portions of the training time some human operators had improperly set up the robot stations. Thus, we decided to not include these results in our paper. We were unable to retry this experiment due to time constraints.

**Real-World Aloha Single-Insertion**    We recreated our simulated Aloha Single-Insertion task in the real world by 3D printing identical assets, and collecting an imitation dataset using Aloha teleoperation. We then trained the Stage 1 (SFT) model with an identical setup (Appendix D.3) and tokenization (Appendix E.2) as the simulated Aloha task. We also built a second variant of our real-world Self-Improvement infrastructure better suited to higher rate control (Appendix N.2). We verified that our Stage 1 BC policies demonstrated reasonable success rates, but we were unable to complete a full Self-Improvement experiment before we had to conclude our project.

## N   Infrastructure Overview

Below we discuss two variations of the infrastructure we implemented for online Self-Improvement. All experiments presented in this work were conducted using Version 1, which proved effective for both simulated and real-world environments operating at control frequencies up to 5Hz.

However, when we attempted to extend our approach to the Real-World Aloha Single-Insertion task (Appendix M), we discovered that the remote action inference server design of Version 1 could not reliably support the 10Hz control frequency required by the Aloha platform. This limitation motivated the development of Version 2, which performs policy inference locally on the actor machines to eliminate network latency. We successfully verified that our Stage 1 behavioral cloning policies performed well on the real robot and briefly tested the Version 2 infrastructure, but time constraints

prevented us from completing a full Self-Improvement experiment for the Aloha Single-Insertion task in the real world.

### N.1    Version 1: Non-Local Policy

Our first infrastructure implementation for Stage 2 Self-Improvement employs a distributed architecture designed to handle both simulated and real-world robot environments. This version separates the computational components from the actors (robots) and uses a client-server model for inference. The system comprises the following key components:

**Dual-Purpose TPU Nodes For Learning and Action Inference**    The same set of TPUs serves dual functionality, alternating between serving as learner nodes and action inference servers depending on the current phase of the RL iteration. During the data collection phase, these TPUs operate as action inference servers, processing observation-goal pairs $(o_t, g)$ and returning actions $a_t \sim p_{\text{action}}^{\text{EFM}}(a_t|o_t, g)$. During the learning phase, they switch to performing policy updates using the data from the replay buffer.

**Steps-to-Go Inference Nodes**    A separate set of inference nodes is dedicated to computing the steps-to-go predictions $d(o, g)$ from Equation 1. These nodes host frozen Stage 1 checkpoints and respond to queries with steps-to-go predictions. The recipients of these predictions then use them for two purposes: (1) computing rewards using Equation 2 by taking differences between consecutive predictions for episode labeling, or (2) evaluating the success condition $\text{success}(o, g) := \mathbb{1}[d(o, g) \leq s]$ during episode execution to determine termination.

**Actors and Environment Loop**    The actors, which can be either real-world robot stations or simulated environments, execute the following environment loop. At the start of each episode, a goal instruction $g$ is sampled. The actor then iteratively: (1) obtains the current observation $o_t$ from the environment, (2) queries the action inference server to obtain action $a_t$, and (3) executes the action in the environment.

In parallel, a background process continuously queries the steps-to-go inference nodes with the most recent observation to check the success condition. This asynchronous design accounts for inference latency—the success condition may not be evaluated for every single observation due to query time, but rather checks the most recent available observation. The episode terminates when either the success threshold is met, the maximum episode length is reached, or a human operator manually intervenes (particularly important for real-world experiments).

Once an episode completes, still within the environment loop, each observation is labeled with its steps-to-go prediction by querying the steps-to-go inference nodes. These predictions are used to compute rewards according to Equation 2 by taking differences between consecutive predictions: $r(o_t, a_t, o_{t+1}, g) = d(o_t, g) - d(o_{t+1}, g)$. The Monte Carlo returns $R_t = \sum_{i=t}^{T} \gamma^{i-t} \cdot r(o_i, a_i, o_{i+1}, g)$ are then calculated, and the labeled tuples $(o_t, a_t, g, R_t)$ are sent to the replay buffer server.

**Replay Buffer Server**    The replay buffer is implemented as a standalone server using Google DeepMind's Reverb [10], which provides efficient distributed data storage and sampling. This server receives labeled experience tuples from all actors and maintains them for training. The learner nodes sample minibatches from this server during the learning phase.

**Central Coordinator**    A central controller orchestrates the entire system, monitoring the replay buffer server's size and coordinating phase transitions. When the replay buffer accumulates sufficient data for $N$ policy updates with batch size $B$ (i.e., $N \times B$ samples), the controller signals all actors to pause data collection and instructs the TPU nodes to switch from inference mode to learning mode. After completing the $N$ RL policy updates, the replay buffer is cleared, and the system transitions back to data collection with the updated policy.

### N.2    Version 2: Local Policy

Our second infrastructure implementation maintains the same overall architecture as Version 1, with one critical difference: policy inference is performed locally on the actor machines rather than

through remote inference servers. This design reduces network latency during episode execution and decouples the learning infrastructure from the inference workload.

**Key Architectural Change** In this version, the TPU nodes are dedicated exclusively to learning. After completing $N$ RL policy updates, the updated model weights are distributed to all actor machines, where they are loaded for local inference. Each actor machine maintains its own copy of the policy model and performs action inference locally during data collection.

**Actor Machines with Local Inference** The actors now handle both environment interaction and policy inference. During episode execution, when an action is needed, the actor queries its local policy model rather than making a network request to a remote server. This eliminates the latency associated with network communication for action inference, which can be particularly beneficial in real-world robotics settings where low-latency control is critical. The background process for success detection remains unchanged—it continues to query the steps-to-go inference nodes with the most recent observation to check the success condition and determine episode termination. Similarly, after episode completion, the actors still query the steps-to-go inference nodes to label each observation for reward computation, exactly as in Version 1.

**Weight Distribution** After each learning iteration, the central coordinator instructs the learner nodes to broadcast the updated model weights to all actor machines. This weight synchronization ensures that all actors use the same policy version during the subsequent data collection phase. The actors pause briefly to load the new weights before resuming episode collection.

**Remaining Infrastructure** All other components remain identical to Version 1:

- The steps-to-go inference nodes continue to operate as separate servers, providing predictions for reward computation and success detection
- The replay buffer server, implemented using Google DeepMind's Reverb [10], continues to centrally manage experience storage and sampling
- The central coordinator maintains its role in orchestrating phase transitions and monitoring the replay buffer size
- Episode labeling with rewards and Monte Carlo returns computation remains unchanged

This local inference approach trades increased memory usage on actor machines (each must hold the full policy model) for reduced inference latency and decreased network traffic during data collection. The design is particularly advantageous when actor machines have sufficient computational resources or when network reliability is a concern.

## O   Broader Impacts

In this work we present a novel Self-Improvement approach that enables Embodied Foundation Models to sample-efficiently improve their task performance, as well as autonomously practice and acquire novel skills. Enabling autonomous Self-Improvement may have significant positive and negative societal impacts that should be carefully considered.

**Potential Positive Impacts** Our approach fundamentally improves policy success rates and can unlock new capabilities. This enables robotics to be applicable in new domains and industries that previously lacked robotic solutions. We also demonstrated that the combination of SFT and Self-Improvement is significantly more sample-efficient that supervised learning alone. This sample-efficiency directly translates to more effective use of resources.

**Potential Negative Impacts** More effective robotic solutions may lead to job loss. Thus, careful attention to opportunity creation is key for the long term stability of society. Enabling robots to more efficiently acquire a broader skillset may also enable unacceptable applications of robots by bad actors.

