# OpenReview forum: "Self-Improving Embodied Foundation Models"
_NeurIPS.cc/2025/Conference — NeurIPS 2025 poster_

### Official Review · Reviewer_DXir · 2025-06-16

**Clarity:** 3
**Significance:** 3
**Originality:** 3
**Rating:** 5
**Confidence:** 4

**Summary:**

The paper presents a two-stage approach for fine-tuning a vision language model to generate actions for robotic applications. In the first stage, the VLM is fine-tuned using both a behavioral cloning loss and a step-to-go loss. The BC loss aims at cloning actions in the trajectories while the step-to-go loss predicts the number of steps required to solve the task (i.e., steps to reach the goal). The underlying assumption is that each trajectory terminates in a goal. The second phase leverages policy gradient (in particular REINFORCE) on top of a reward function based on the step-to-go prediction to improve task success using online interactions. The authors showed that this approach is sample efficient and leads to significant performance improvements over the two considered domains.

**Questions:**

- My main concern is about the dataset assumption "all episodes in the dataset end in a state where the episode goal is accomplished". This seems quite a strong assumption. Could you clarify what your meaning of a goal is? In principle, every state is a goal. Are you simply considering that if a trajectory ends in goal g, the intent of the trajectory was to reach goal g? Or do you want something stronger, such as that you cannot have failures in the dataset? If so, why do you need that? How could you remove such assumption?

- Another main limitation is that this work is specifically designed for goal-reaching tasks. How would you extend your approach beyond this class of problems?

- Modern VLA has moved from action tokenization to direct prediction (e.g., using flow-matching). Any reason for using action tokenization in this work? What do you expect to change in that case?

- Do you test multiple realizations of the 10%, 20% and 80% datasets or you consider only one realization? You mentioned that you considered 3 random seeds, could you explain the protocol? Did you repeat each phase 3 times? I could not find the unaggregate results in appendix. Could you add standard deviation (or any other metric) in Figure 3?

- Figure 2, could you make clear what is the value of the predicted steps-to-go in moment 1? It seems to be very close to the condition `d<=3` that you use to terminate the episode? This makes me wonder that a slightly wrong hyper-parameter may result in an overall failure of the approach.

- Appendix needs quite a bit of work. There are several sections not finished. Double check the main paper for missing references (e.g. in line 98).

- Could you clarify the input to the VLM? Do you use a single image? Do you stack images to have information about velocity, etc? Do you use also state-based information?

**Ethical Concerns:**

["NO or VERY MINOR ethics concerns only"]

**Final Justification:**

The authors addressed my concerns and I decided to increase the score. I think the paper is novel and interesting for the community.

**Limitations:**

Yes

**Quality:**

3

**Strengths And Weaknesses:**

Strengths
- Overall I found the paper quite interesting and ready to read
- The approach is interesting and results are promising

Weaknesses
- See section below

---

> ### Author Rebuttal · Authors · 2025-07-31
>
> Thank you for your detailed and insightful review of our work! We hope our responses below address your questions. We look forward to a productive author-reviewer discussion period!
>
> **Goals & Dataset Assumption**\
> Thank you for this insight! Indeed we meant that “if a trajectory ends in goal $g$, the intent of the trajectory was to reach goal $g$”, with no further assumptions. This definition, however, only captures a subset of the settings we can address, and in fact our approach is more broadly applicable. More carefully stated, what we care about is that we can sample tuples $(o_t, a_t, g, t_g)$ where $o_t$ is the observation, $a_t$ is the associated action in the dataset, $g$ is a future goal that is accomplished at timestep $t_g$ in the trajectory where $(o_t, a_t)$ appeared. This statement subsumes the domains we considered in our work (LanguageTable with language goals, as well as the single-task domains), while incorporating other scenarios such as the one you pointed out, “In principle, every state is a goal”. We will make this update to our main manuscript.
>
> **Non-Goal-Reaching Tasks**\
> In this work we did not consider non-goal-reaching tasks. One potential avenue towards addressing such domains may come from literature connecting discounted RL in MDPs with Stochastic Shortest Paths (SSP). Specifically, there exists a construction that reduces any $γ$-discounted, bounded-reward MDP into an SSP [1]. Since SSP is a goal-reaching task with reward 1 in the goal state and 0 elsewhere, we may be able to apply our proposed approach.
>
> **Action Prediction**\
> Our approach does not make any particular assumptions about the action prediction approach. One consideration when leveraging diffusion/flow-matching policies is that due to intractable log-likelihood, we would need to replace on-policy REINFORCE with a different RL algorithm. Recently there has been an explosion of RL methods designed for such policy formulations. More broadly, however, our general framework, reward model, and success detector do not need to change.
>
> **Dataset Sizes**\
> We create one realization of each dataset size, then train Stage 1 once for each dataset size, and subsequently train Stage 2 with 3 random seeds. The unaggregated results appear in Appendix H, Figure 10, Left. We will update Figure 3 with error bars.
>
> **Success Detection Threshold**\
> In simulated domains since we have access to the simulation state, we do not use the threshold success detector. We only use thresholding success detection for real-world experiments. Thus moment 1 in Figure 2 would not trigger the success condition. For each real-world domain, we decided the threshold by visualizing model predictions on various trajectories. The videos in the LanguageTable section of our supplementary videos website (link at the end of Section 1 of our manuscript) visualize many real-world trajectories while plotting the expected value of steps-to-go across time. As can be seen, the thresholding mechanism is quite reliable.
>
> Furthermore, so long as the threshold value is not too high (i.e. not too lenient) it would not have a large detrimental impact. If the threshold is too low, the main effect is that the episodes will contain a bit more amount of states near the goal.
>
> **Appendix Cleanup**\
> Thank you for the note! In Appendix K we plan on including a diagram depicting the infrastructure we built for on-robot on-policy RL with large foundation models.
>
> **Observations**\
> For the LanguageTable and BananaTable domains we include the current and previous image frames, and the input text represents the goal instruction. The output is a discretized action for the x direction and a discretized action for the y direction.
>
> In Aloha, there are 4 cameras, but to mitigate training compute requirements, we a) stack the top and table view images into one image, and b) stack the left and right wrist view into one image as well, and pass these two images to the PaLI model. The input text includes discretized joint angle information. The output is an action chunk of 5 Aloha actions, with each dimension of each action being represented by a single token. Thus the output is 5 x 14 tokens.
>
> For details we invite you to refer to Appendices D & E.
>
> [1] https://www.cse.iitm.ac.in/~prashla/notes/rlbook/discounted-cost-mdps.html#equivalence-of-discounted-mdp-and-ssp

---

> > ### Comment · Reviewer_DXir · 2025-08-05
> >
> > Thank you for the clarification, most of my concerns have been addressed. I think all these comments have to be integrated into the revised version of the paper.

---

> > > ### Author Response · Authors · 2025-08-08
> > > **Authors' Response**
> > >
> > > We will ensure to integrate the above updates into our manuscript. Thank you for your detailed reviews and engagement in this review process!

---

### Official Review · Reviewer_vuwt · 2025-06-25

**Clarity:** 4
**Significance:** 3
**Originality:** 3
**Rating:** 6
**Confidence:** 4

**Summary:**

The authors propose to fine-tune foundation models for robotics by introducing a RL phase after training, similar to LLMs. In order to obtain a suitable reward function from offline imitation learning, the method trains the foundation model offline on imitation learning data with two objectives:
- A behavior cloning objective for learning actions given state and goal
- A "steps-to-go" objective, which lets the model predict the remaining length of the trajectory until the goal condition is met.

During the reinforcement learning "self-improvement" phase, the steps-to-go prediction serves as a reward function. If an action reduces the number of predicted steps, i.e. is an action towards the goal, the reward is positiv. if it increases the number of predicted steps, the action was counter to the goal and the reward is negativ. This objective can also serve as an automatic success detector for real world RL, when the steps-to-go prediction achieves a sufficiently small value.

The models are evaluated regarding success improvement and data efficiency in simulation and real-robot tasks, namely LanguageTable and Aloha setups.

**Questions:**

## See Weaknesses.

### Additionally, for my understanding:
For the reward function / steps-to-go prediction. I am wondering about a case, where the robot does something which does not change the steps-to-go prediction, e.g., moving on a circle around an object. The reward signal in this case would be 0. Would it make sense to include a small penalty for such cases, for example $r = d_t - d_{t+1} - 0.1$ ?

Figure 2, step 15 (2nd red circle) seems to be close to this edge case.

**Ethical Concerns:**

["NO or VERY MINOR ethics concerns only"]

**Final Justification:**

I appreciate the authors detailed responses in the discussion phase. I am convinced that this is elegant work with potentially high impact for robot learning.

**Limitations:**

yes

**Quality:**

4

**Strengths And Weaknesses:**

## Strength
- Elegant reward function / model design
- solid theoretical analysis
- experiments include both sim and real-robot and support the claims regarding data, human-effort efficiency and data generalization capabilities
- well written

## Weakness
- as mentioned in the limitations, the current experimental evaluation is on relatively simple tasks when compared to CALVIN or LIBERO benchmarks, which require long horizon manipulation, chaining of multiple skills and handling of distractor objects or "unseen" objects hidden inside a desk drawer. It is unclear how this approach would perform in these settings.
- The aloha peg insertion experiment requires an adaptation of the reward function, as the steps-to-go function is unable to detect a successful peg insertion from cameras alone. I believe this is an important limitation which should be communicated more clearly. Additionally, it is unclear to me how the success is detected to add the constant reward.

### Minor Comment
- I believe Figure 3 could be easier to read, if all the subplots would use the same Y-Axis scaling.

---

> ### Author Rebuttal · Authors · 2025-07-31
>
> Thank you for your detailed and insightful review of our work! We hope our responses below address your questions. We look forward to a productive author-reviewer discussion period!
>
> **Skill-Chaining**\
> The focus of our work is to introduce a very effective framework for post-training embodied foundation models for a downstream skill. We did not consider tasks comprised of a sequence of subskills.
>
> As discussed in our Future Works section, we believe our success detector based on steps-to-go prediction can be effectively leveraged to address skill-chaining and long-horizon tasks: during rollouts, once the success detector indicates that a sub-task is completed, the policy moves onto the next sub-task. This extension of our approach can enable simultaneous RL training on all the subskills needed for a desired long-horizon task.
>
> **Unseen Objects**\
> In our work we present a novel combination of web-scale pretrained foundation models + an effective on-robot online RL post-training approach. We believe that this combination can address unseen objects in two ways:
>
> **Semantic:** When the difference between objects is mostly perceptual and semantic (i.e. similar physical properties), the generalization inherited from the foundation models can address the gap. This was indeed observed in RT-2 [1] which trained robot policies through behavioral cloning on top of a pretrained vision-language foundation model (equivalent to our Stage 1 policy).
>
> **Novel Behavior:** The more challenging approach is learning to handle unseen objects which also have unseen properties. This is one of the key contributions of our work. Our approach uniquely unlocks this form of generalization as observed in our BananaTable experiments (Section 4.3.2). The mechanism that enables this generalization is as follows:
> Before Stage 1, the models are initialized from a PaLI vision-language model (VLM) that is pretrained on the web-scale data
> After Stage 1 on LanguageTable data, the reward model understands what the task of “move the X to the Y” means for the blocks, but because it starts from the PaLI VLM, it generalizes its understanding of “move the X to the Y” from blocks to arbitrary objects such as bananas.
> During Post-Training, this enables to policy to practice manipulating the banana and learning how deal with the physics of this new object’s properties
>
> **Aloha Peg Success**\
> Thank you for this note. We will update the manuscript to more clearly communicate this change. As to the specifics of success detection in this setting, since this was a simulation domain we were able to detect the success condition from the simulation state.
>
> We would also like to provide some additional context as to why this success signal was needed. We had designed this task to trigger the success when the peg was fully pushed into socket. Specifically, the mesh of the peg would have to come into contact with the inside wall at the end of the socket. Without the success bonus, the policy would still learn to successfully accomplish the peg insertion task, however, it would not try to push the peg all the way back to the end of the socket, and therefore not trigger the success. Upon closer inspection we saw that with the Aloha camera views, and the image resolutions we were using, it was extremely difficult to tell the difference between when the peg was fully pushed in vs. “almost fully” pushed in. Hence the reward model would not be able to incentivize fully pushing the peg in. The success bonus made it so that the policy would learn to push the peg further and trigger the success mechanism.
>
> In addition to communicating the success bonus more clearly in the main text, we will include these notes in our Appendix as well.
>
> **Reward Signal 0**\
> It is correct that if the robot executes an action that does not change the reward function the reward would be 0. However, if the robot had executed a better action and made progress towards the goal, it would have received a positive reward (and conversely a worse move would have received a negative reward). Thus the robot is still incentivized to choose actions that make progress towards the desired goal.
>
> [1] RT-2: Vision-Language-Action Models Transfer Web Knowledge to Robotic Control

---

> ### Comment · Reviewer_vuwt · 2025-08-02
>
> Thank you for your detailed response and helping me better understand your impressive work. Especially the added information for the ALOHA experiment is much appreciated.

---

> > ### Author Response · Authors · 2025-08-08
> > **Authors' Response**
> >
> > Thank you for your detailed reviews and engagement in this review process!

---

### Official Review · Reviewer_E1GX · 2025-06-29

**Clarity:** 2
**Significance:** 3
**Originality:** 2
**Rating:** 4
**Confidence:** 4

**Summary:**

This paper proposes a two-stage post-training pipeline for embodied foundation models, using a steps-to-go reward which represents the model's estimate of how many time steps remain until task completion and serves as an implicit measure of progress. First, a supervised fine-tuning stage adapts a pretrained vision-language backbone model with a dual loss that combines behavioral cloning and steps-to-go prediction. Then, the trained steps-to-go head is frozen, and its output is reused as both a shaped reward and a success detector, enabling efficient on-policy reinforcement learning on real or simulated robots with minimal human oversight.

**Questions:**

- How does the method account for diversity in action trajectories that achieve the same semantic goal but follow different physical paths? For example, consider a task where the robot must move an object from point A to point B. One trajectory might approach the goal directly in a straight line, while another might take a longer but equally valid path around an obstacle or due to environmental constraints. Since the steps-to-go reward is based on minimizing the number of steps, does the method implicitly favor shorter trajectories? If so, could this discourage exploration and lead to a collapse toward only the shortest, most direct behaviors, thereby reducing the diversity and robustness of learned policies?
- If more computational resources were available, would it be beneficial to freeze the entire Stage 1 checkpoint (not just the steps-to-go head) during Stage 2 RL fine-tuning? Could this help stabilize training or improve sample efficiency?

**Ethical Concerns:**

["NO or VERY MINOR ethics concerns only"]

**Final Justification:**

Most of my concerns have now been resolved, and I appreciate how clearly authors articulated the key contributions. In particular, authors detailed explanation regarding the design of a general-purpose reward model, which minimizes the need for task-specific engineering, and interpretation of the experimental videos were convincing enough to change evaluation.

**Limitations:**

Yes, the authors have adequately addressed the limitations of their work. However, it would be beneficial to clearly discuss scenarios where the application of the steps-to-go reward might be challenging.

**Paper Formatting Concerns:**

Figures 1, 2, 3, and 5 overflow the text linewidth and should be adjusted for proper formatting.

In Section 2.2 (Reward Function paragraph), a section reference appears to be missing and should be added for clarity.

**Quality:**

2

**Strengths And Weaknesses:**

> ### Strenghts
- The proposed steps-to-go reward is conceptually simple yet effective.
- The framework is validated across multiple environments: Language Table (real-world and simulator), Banana Table (real-world), and Aloha single insertion.
- Figure 2 provides a clear visual demonstration of how steps-to-go functions in real environments, enhancing intuitiveness.
- In Appendix, The framework demonstrates compatibility with other vision-language backbones, suggesting broader applicability.

---

> ### Weaknesses
- The primary contribution of the paper lies in demonstrating the efficacy of the steps-to-go reward design. However the paper lacks thorough comparison or discussion with other established reward design approaches.
- Prior work on mitigating sparse rewards, such as subgoal-based methods or goal-state similarity-based reward relabeling, is not sufficiently acknowledged or contrasted. A deeper analysis would help clarify when and why steps-to-go provides a significant advantage.
- The generalization capability of the method may be sensitive to the sampling frequency of robot’s trajectory data. For instance, even for trajectories that perform the same behavior, the number of steps required to reach the goal can vary depending on the frame rate. In practical settings involving mixed sampling frequencies (for example, 60 fps and 30 fps), the paper provides limited discussion on how steps-to-go predictions are normalized or calibrated. This raises concerns about the scalability and robustness of the framework.
- The novel skill acquisition results in Section 4.3.2 are intriguing but may be slightly overstated. The instruction format “move the X to the Y” is already present in both the Language Table and Banana Table datasets, which weakens the claim of strong generalization to novel skills. Moreover, it remains unclear how the model generalizes to previously unseen object properties.

---

> ### Author Rebuttal · Authors · 2025-07-31
>
> Thank you for your detailed and insightful review of our work! We hope our responses below address your questions. We look forward to a productive author-reviewer discussion period!
>
> **Comparisons to Other Reward Models**\
> Related works, existing reward models fall short in one way or another
> Our reward model is general, very straightforward to incorporate into foundation model training
> Being able to demonstrate real-world results in such a setting is an incredible undertaking, requiring massive infrastructure efforts. Thus the fact that such an approach work so well is a very significant for the robotics community.
>
> **Comparisons to Other Reward Models & Discussion of Prior Works**\
> Our objective in designing our reward model was to have a) a very general purpose reward model that requires minimal task-specific engineering, and b) is very straightforward to incorporate into large scale training of foundation models. Existing reward models fall short in one way or another. Due to space limitation we present a condensed form of our Related Works discussion in the main manuscript, but provided a much more extensive discussion in Appendix L. Specifically for the forms of reward modeling you mentioned:
> Lines 1144-1146: we discussed goal-state similarity-based reward
> Lines 1151-1152: we discussed an example goal relabeling based method
> Lines 1158-1167: we discussed timestep distance based methods
>
> We would like to also emphasize that being able to demonstrate 1) real-world robotics results, with 2) foundation models, 3) trained with on-robot online RL, 4) with a general-purpose learned reward, is a very large undertaking that pushes the boundaries of current robotics methodologies. We believe that demonstrating the fact that such an approach can result in such significant improvements, sample-efficiency, and generalization is a very significant contribution to the robotics community in its own right.
>
> **Mixed Sampling Frequencies**\
> In this work we did not study mixed sampling frequencies as it is not a standard variation that is typically studies, but it could indeed by a realistic setting that may occur when training multi-embodiment foundation models. Our initial idea for how to address this setting is to perform steps-to-go prediction in terms of wall-clock-time, instead of “environment steps” which would be FPS-dependent.
>
> **Novel Skill Acquisition**\
> During post-training the policy learns to significantly update its motor primitive and learn how to effectively manipulate a banana. The success rate improvement from 63% to 85% (Section 4.3.2), on its own, does not sufficiently capture the extent to which the policies become better at this task. We strongly encourage viewing the videos in our supplementary videos website (link at the end of Section 1 of our manuscript).
>
> The video of the policy before post-training shows that the policy is very inefficient, often making the banana rotate around itself instead of moving in the intended direction. The video after post-training, shows that the policy has learned how to deal with the novel physics of the banana, and steadily push the banana in the right direction.
>
> The mechanism that enables this generalization is as follows:
> Before Stage 1, the models are initialize from a PaLI vision-language model (VLM) that is pretrained on the web-scale data
> After Stage 1 on LanguageTable data, the reward model understands what the tasking of “move the X to the Y” means for the blocks, but because it starts from the PaLI VLM, it generalizes its understanding of “move the X to the Y” from blocks to arbitrary objects such as bananas.
> During Post-Training, this enables to policy to practice manipulating the banana and learning how deal with the physics of this new object’s properties
> This is one of the key contributions of our work. This result is uniquely unlocked by our proposed novel combination of pretrained foundation mode + post-training approach.
>
> **Diversity in Action Trajectories**\
> In our work, we are learning the **distribution** over steps-to-go which is very multimodal due to mistakes, and many different ways of accomplishing the same goal:
> In LanguageTable the human demonstrations are not perfectly efficient and often make mistakes
> In the Aloha data there are mistakes such as a) peg being dropped, b) a socket being dropped, c) accidental drops or mistakes during insertion.earning steps-to-go from such data is in fact a **significant feature** of our work. This enables the model to learn the **distribution** over steps-to-go.
>
> In our supplementary videos website (link at the end of Section 1 of our manuscript), the top 2 videos in the **Aloha** section demonstrate this ability to learn the multimodal distribution. In these videos:
> Top: Shows the camera views
> Middle plot: Expectation of the steps to go distribution across time. x-axis is time, y-axis is expected value of steps-to-go prediction.
> Bottom plot: Shows the predicted probability distribution over steps-to-go at that moment in time. x-axis is steps-to-go, y-axis is probability.
> We encourage you to focus on the bottom plot while viewing the videos. You can observe that at critical times (e.g. about to pick a peg, about to do an insertion, etc.) the model makes very multimodal predictions, and once the outcome is clear (e.g. successful picked up the peg) the prediction become more confident and uni-modal.
>
> Thus from the perspective of learning the multimodal steps-to-go distribution, our method in its current form already accomplishes this desiderata. Learning steps-to-go from such data is in fact a **significant feature** of our work. As discussed in Section 2.2, learning the distribution over steps-to-go opens up many avenues of future work such as learning risk-aware policies using CVaR, or distributional reinforcement learning.
>
> We believe the second component to your question, “does the method implicitly favor shorter trajectories” is independent of our proposed approach and relates to the specific choice of RL algorithm that is used for optimizing the reward function. In our work we used on-policy REINFORCE which will favor shorter trajectories. In contrast, if for example we decided to use Maximum Entropy RL (Max-Ent RL) [1], we could tune the extent to which the policy optimizes for shorter trajectories: In Max-Ent RL, the goal is to learn a policy whose trajectories $\tau$ have the probability distribution $$p(\tau) \propto \exp\left(\frac{\text{returns}}{\text{temperature}}\right)$$. Thus by tuning the temperature parameter, we can tune the extent to which the policy cares about optimizing for shorter trajectories.
>
> **Freezing Entire Stage 1 Checkpoint**\
> Would you be able to clarify your question please? During Stage 2 we have one frozen checkpoint acting as reward model and success detector, and a non-frozen checkpoint which is the policy and is being updated during Stage 2. If we freeze everything, which component would be updated during Stage 2 training to improve the policy?
>
> **Limitations**\
> In Appendix M we have provided an extensive discussion on Limitation and avenues of Future Work, discussing important challenging cases that we anticipate.
>
> [1] Reinforcement Learning and Control as Probabilistic Inference: Tutorial and Review

---

> > ### Comment · Reviewer_E1GX · 2025-08-01
> >
> > Thank you for the clarification and for thoroughly addressing my questions. Most of my concerns have now been resolved, and I appreciate how clearly you articulated the key contributions. In particular, your detailed explanation regarding the design of a general-purpose reward model, which minimizes the need for task-specific engineering, and your interpretation of the experimental videos were convincing enough to change my evaluation.
> >
> > Regarding my follow-up to Question 2, the reason I asked was to better understand the effectiveness of the proposed method in a setup where the Stage 1 checkpoint is frozen and other adapters, such as LoRA, are applied.
> >
> > * In Stage 2, had adaptation methods such as LoRA already been applied during model training?
> > * During Stage 2, do the model parameters undergo significant changes? I would be particularly interested in the authors’ insights on this point. Even preliminary thoughts or minor observations would be very helpful.

---

> ### Author Response · Authors · 2025-08-08
> **Authors' Response**
>
> Thank you for your response, and for the clarification on Question 2.
>
> Indeed, we believe that alternative approaches for Stage 2 fine-tuning, such as LoRA can be very valuable. As an example, we can consider a scenario where a "Base Model" is trained in Stage 1 and we have many different downstream tasks of interest. One way to address this demand is to perform separate Stage 2 fine-tunes for different downstream tasks using LoRA. Consequently, we would significantly reduce the computational demand for Stage 2 finetunes ($N$ LoRA finetunes instead of $N$ Base Model finetunes). We will add this discussion to our Future Works section.
>
> As described in our manuscript, during Stage 2 we froze the vision component of the Vision Language Model (VLM) and only updated the parameters of the language Transformer component. This was a very early decision in our project that was made in hopes of training stability and reduction in computational demand for Stage 2. Thus, any parameter updates during Stage 2 a limited to this component.
>
> One indicator we can use for how much the parameters changed during Stage 2 fine-tuning is the number of gradient steps. Across the board, in our experiments we showed that Stage 2 fine-tuning is very sample-efficient in terms of amount of environment interactions. Since we used on-policy REINFORCE without data reuse, this also meant that we did very little gradient steps in comparison to Stage 1 training. More concretely, while Stage 1 performed on the order of O(100K) gradient steps until it reached the best validation losses, Stage 2 fine-tuning only performed O(1K) gradient steps until it reached the peak success rates. Thus, we are inclined to believe that parameters did not change very significantly during Stage 2 fine-tuning, and that the Self-Improvement process rapidly made fine adjustments to the parameters to make the policy achieve high success rates.
>
> Thank you for motivating us to consider this aspect. We will update our manuscript to highlight that the combination of SFT + Self-Improvement is significantly more sample-efficient that SFT alone in three key respects:
> 1) Amount of robot-environment interaction needed
> 2) Amount of human effort needed (1 human monitoring multiple robots vs. many humans collecting data on many robots)
> 3) Amount of training steps needed, which also significantly reduces the amount of computation resource demands
>
> Lastly, very recent work in the LLM domain suggests that RL fine-tuning results in much sparser updates to model parameters in comparison to SFT [1]. Notably, they mention, "RL leads to consistently sparse parameter updates (often >70% sparsity) while SFT produces dense updates. Sparsity emerges as a characteristic property of RL finetuning in LLMs." Such observations may go hand-in-hand with our discussion above regarding the significant efficiency of Stage 2 in terms of number of gradient updates.
>
> Again, thank you for your detailed reviews and engagement in this review process!
>
> [1] "Reinforcement Learning Finetunes Small Subnetworks in Large Language Models" https://arxiv.org/abs/2505.11711

---

### Official Review · Reviewer_UpTX · 2025-07-03

**Clarity:** 2
**Significance:** 3
**Originality:** 2
**Rating:** 4
**Confidence:** 3

**Summary:**

This paper presents a two-stage framework for post-training embodied foundation models (EFMs): (1) Supervised Fine-Tuning (SFT) using behavioral cloning and steps-to-go prediction; and (2) a novel Self-Improvement phase that leverages the learned steps-to-go estimates to define a shaped reward function and a robust success detector for online reinforcement learning. The method is tested on two robot platforms—LanguageTable and Aloha—both in simulation and real-world environments. The authors show significant gains in sample efficiency, performance robustness, and skill generalization beyond imitation data. Notably, the method enables autonomous acquisition of novel behaviors such as BananaTable manipulation.

**Questions:**

1.	Have you explored off-policy RL algorithms (e.g., SAC, TD3) using the same steps-to-go reward function? What challenges did you face?

2.	How robust is the success detector threshold (e.g., s = 3) across domains or tasks? Could an adaptive threshold further improve learning?

3.	Is the PaLI reward model fine-tuned or frozen during RL training? Would co-training improve performance or degrade stability?

4.	In the BananaTable task, does the agent develop new motor primitives or just repurpose existing skills in novel contexts?

**Minors:**

-	Line 98: in RL (Section ??).

**Ethical Concerns:**

["NO or VERY MINOR ethics concerns only"]

**Final Justification:**

The author's rebuttal has addressed most of my concerns.

**Limitations:**

1.	The method depends on on-policy RL with no data reuse, which may limit scalability to more complex tasks.

2.	The frozen reward model from SFT assumes prediction accuracy remains consistent across domains; this may fail under strong domain shifts.

3.	The steps-to-go abstraction implicitly assumes task progress can be estimated linearly or monotonically, which may not hold in deceptive or delayed-reward settings.

4.	While BananaTable and insertion tasks are insightful, broader application across locomotion, dexterous manipulation, or non-prehensile control would strengthen generality claims.

**Quality:**

3

**Strengths And Weaknesses:**

**Strengths**:

- The paper presents a clean and conceptually grounded two-stage pipeline for post-training embodied foundation models (EFMs). By combining supervised fine-tuning (SFT) with a self-supervised online reinforcement learning phase (Self-Improvement), the authors draw a compelling parallel to the LLM post-training recipe (SFT + RLHF). This adaptation to robotics is well-motivated and could establish a new foundation for scalable robot policy training without explicit reward functions.

- The proposed reward shaping mechanism—using the decrease in predicted steps-to-go as a proxy for task progress—is both elegant and practical. It removes the need for manually engineered rewards or hand-crafted success detectors. The success indicator derived from steps-to-go thresholding (e.g., d(o, g) ≤ s) is a clever way to detect task completion robustly in real-world conditions where ground-truth success metrics are often unavailable.

- The experiments are impressively comprehensive, spanning simulation and real-world settings, two distinct robot platforms (LanguageTable and Aloha), and a diverse range of training set sizes and generalization scenarios. The authors go beyond mere benchmarks: they conduct ablations on reward model quality (PaLI vs. Uni-PaLI vs. Scratch), show robustness across random seeds, and highlight domain transfer through Real2Sim and BananaTable evaluations.

**Weaknesses**:

- The reinforcement learning component relies on a straightforward on-policy REINFORCE algorithm. While this choice offers training stability and avoids the need for value estimation, it misses out on potential gains in sample efficiency from off-policy algorithms (e.g., SAC, TD3, or even Q-learning variants). The absence of any baseline comparisons using such alternatives limits our understanding of the true potential of the proposed reward signal under modern RL schemes.

- The method is primarily empirical, with no theoretical analysis of convergence, stability, or generalization guarantees. For example, the steps-to-go reward assumes monotonic progress toward the goal, but this may not hold in multi-modal or deceptive tasks. More discussion on the theoretical assumptions behind using steps-to-go as a proxy for value could strengthen the paper’s foundation.

- While the BananaTable task offers strong qualitative evidence of generalization, the paper lacks formal metrics or benchmarks to quantify generalization gaps (e.g., success rates under zero-shot task variations, trajectory diversity metrics, or skill composition tests). The evaluation focuses on success rate improvements within the same task class, which may not fully capture the policy’s ability to generalize to novel skill compositions or long-horizon dependencies.

---

> ### Author Rebuttal · Authors · 2025-07-31
>
> Thank you for your detailed and insightful review of our work! We hope our responses below address your questions. We look forward to a productive author-reviewer discussion period!
>
> **On-Policy REINFORCE**\
> Our work introduces many novel components into a full framework for post-training embodied foundation models, with emphasis on reliability in order to be successful in real-world experiments. This, in this work from the beginning we used on-policy REINFORCE and did not attempt any off-policy methods. This was motivated by a number of factors:
> - On-Policy methods are known to be more stable in general. On-Policy REINFORCE specifically removes two vertices of the deadly triad: off-policy learning & bootstrapping
> TD-Learning with large models has significant stability issues, and designing stable methods is an active area of research [1]
> The current choices for reinforcement learning with Large Language Models (LLMs) are based on on-policy methods as well (e.g. DeepSeek-R1 [2]). In fact [2] introduced the GRPO on-policy RL algorithm which does not learn a value function either. GRPO is rapidly becoming one of the most popular algorithms used in training LLMs.
>
> The fact that our results demonstrates such significant sample-efficiency with on-policy REINFORCE and no data reuse demonstrates that any advancements in off-policy training methods for large foundation models stands to improve our results.
>
> **Monotonic Progress Assumption & Delayed Rewards Settings**\
> While it is accurate that we do not provide convergence, stability, or generalization guarantees, it is **not the case** that the steps-to-go reward model assumes monotonic progress towards the goal. In our work, we are learning the **distribution** over steps-to-go which is very multimodal due to mistakes, and many different ways of accomplishing the same goal:
> In LanguageTable the human demonstrations are not perfectly efficient and often make mistakes
> In the Aloha data there are mistakes such as a) peg being dropped, b) a socket being dropped, c) accidental drops or mistakes during insertion.earning steps-to-go from such data is in fact a **significant feature** of our work. This enables the model to learn the **distribution** over steps-to-go.
>
> In our supplementary videos website (link at the end of Section 1 of our manuscript), the top 2 videos in the **Aloha** section demonstrate this ability to learn the multimodal distribution. In these videos:
> Top: Shows the camera views
> Middle plot: Expectation of the steps to go distribution across time. x-axis is time, y-axis is expected value of steps-to-go prediction.
> Bottom plot: Shows the predicted probability distribution over steps-to-go at that moment in time. x-axis is steps-to-go, y-axis is probability.
> We encourage you to focus on the bottom plot while viewing the videos. You can observe that at critical times (e.g. about to pick a peg, about to do an insertion, etc.) the model makes very multimodal predictions, and once the outcome is clear (e.g. successful picked up the peg) the prediction become more confident and uni-modal.
>
> Learning steps-to-go from such data is in fact a **significant feature** of our work. As discussed in Section 2.2, learning the distribution over steps-to-go opens up many avenues of future work such as learning risk-aware policies using CVaR, or distributional reinforcement learning.
>
> With respect to “delayed reward settings”, steps-to-go prediction also leads to reward densification. As an example, in the LanguageTable Block2Block task, a standard reward function would the negative L2 distance from block A to block B. However, this reward would not provide any signal which the arm is moving toward block A. Instead, steps-to-go prediction provides dense reward signals that accurately provide signal to the arm moving the correct position to effectively push block A. Videos demonstrating such predictions can be viewed in our supplementary videos website (link at the end of Section 1 of our manuscript) where we plot the expected value of steps-to-go predictions alongside LanguageTable trajectories.
>
> **Skill Composition Generalization**\
> The focus of our work is to introduce a very effective framework for post-training embodied foundation model for a downstream task class. Thus, in the BananaTable task the notion of strong generalization that we focus on is the ability to learn a novel behavior, namely how to handle the physics of a new object. As discussed in Section 4.3.2, in the BananaTable task the success  rate improves from 63% to 85%, and the videos in our supplementary videos website (link at the end of Section 1 of our manuscript) demonstrate that this is a significant generalization gap to bridge, as the BC models are very inefficient at accomplishing this task.
>
> In terms of skill composition and long-horizon tasks, as discussed in our Future Works section, we believe our success detector based on steps-to-go prediction can be effectively leveraged to perform skill-chaining: once the success detector indicates that the sub-task is completed, the policy moves onto the next sub-task.
>
> **Robustness of Success Detector Threshold**\
> We chose this threshold via analyzing the expected value of steps-to-go predictions on a small subset of the dataset. We have not investigated adaptive thresholding.
>
> **Frozen Reward Model**\
> Per our discussion in Section 2.2, we keep the reward model frozen during RL training, mimicking default RL paradigms where reward functions typically do not change during training. For our specific choice of reward model, due to the connection to Value Functions and baselines (Appendix F), there may be interesting connections to value function training and off-policy RL. In turn, updating the reward model might suffer from similar instability issues as value functions (discussed earlier above).
>
> **BananaTable New Skills**\
> During post-training the policy learns to significantly update its motor primitive and learn how to effectively manipulate a banana. The success rate improvement from 63% to 85% (Section 4.3.2), does not sufficiently capture the extent to which the policies become better at this task. We strongly encourage viewing the videos in our supplementary videos website (link at the end of Section 1 of our manuscript). Before post-training the policy is very inefficient, often making the banana rotate about itself instead of moving in the intended direction. After post-training, the policy can much more steadily push the banana in the right direction.
>
> [1] Stop Regressing: Training Value Functions via Classification for Scalable Deep RL
> [2] DeepSeek-R1: Incentivizing Reasoning Capability in LLMs via Reinforcement Learning

---

### Note · Authors · 2025-08-16

We thank the reviewers for their detailed reviews of our work and the AC for their time.

In this work we presented a novel two-stage post-training approach for training Embodied Foundation Models. The first stage, Supervised Fine-Tuning (SFT), fine-tunes pretrained foundation models on imitation learning datasets using both: a) behavioral cloning, and b) steps-to-go prediction objectives. Leveraging the foundation model’s steps-to-go prediction ability, we designed a general-purpose reward function and success detector that eliminates task-specific reward-engineering and instrumentation in the real world. In the second stage, Self-Improvement, our proposed reward function and success detector enable a fleet of robots to autonomously practice downstream tasks with minimal human supervision, leading to rapid improvement through online reinforcement learning. We presented a series of key findings and novel results that significantly impact the future of how foundation models are trained for robotics.

We had an engaged and productive review process with the reviewers. Following the discussion period, reviewers noted that they have minimal lingering concerns. Reviewers agreed with the key novel contributions and results in our work. Additionally:
- Reviewers agreed that our approach could significantly impact the future of how general-purpose embodied foundation models are trained. In particular we emphasize the following quote: “...the authors draw a compelling parallel to the LLM post-training recipe (SFT + RLHF). This adaptation to robotics is well-motivated and could establish a new foundation for scalable robot policy training without explicit reward functions.”
- They agreed that our proposed reward and success detection approach based steps-to-go prediction are a) elegant and practical, b) remove the need for task-specific engineering, c) valuable for real-world robotics since ground-truth rewards and success detection are not available.
- Reviewers noted that our experiments are extensive and comprehensive, covering real-world, simulation, multiple robotic platforms, diverse data settings, ablations, and multiple experiments designed for evaluating generalization capabilities.
- Reviewers appreciated the generality of our approach, and how it can be readily applied to any pretrained foundation model backbone.
- Multiple reviewers noted the clarity of our manuscript.

We appreciate the constructive engagement from the reviewers and the AC. Thank you!

---

### Decision · Program_Chairs · 2025-09-17

**Decision:**

Accept (poster)

**Comment:**

This paper develops a new two-stage framework for post-training embodied foundation models: supervised fine-tuning with behavioral cloning and steps-to-go prediction, followed by a self-improvement phase that uses the learned steps-to-go estimates to shape rewards and detect task success for online reinforcement learning. Tested on LanguageTable and Aloha robots in simulation and the real world, the method improves sample efficiency, robustness, and skill generalization beyond imitation data, and enables autonomous learning of novel behaviors such as BananaTable manipulation. It reached a consensus that the approach proposed is novel, conceptually grounded and simple, and the experiments are comprehensive. The overall writing and presentation are also clear. I suggest the authors incorporate the feedback from the reviewers in preparing the camera-ready version.